# An actin-related protein that is most highly expressed in *Drosophila* testes is critical for embryonic development

Courtney M Schroeder[1]*, Sarah A Tomlin[1,2], Isabel Mejia Natividad[1,2], John R Valenzuela[1], Janet M Young[1], Harmit S Malik[1,2]

[1]Division of Basic Sciences, Fred Hutchinson Cancer Research Center, Seattle, United States; [2]Howard Hughes Medical Institute, Fred Hutchinson Cancer Research Center, Seattle, United States

**Abstract** Most actin-related proteins (Arps) are highly conserved and carry out well-defined cellular functions in eukaryotes. However, many lineages like *Drosophila* and mammals encode divergent non-canonical Arps whose roles remain unknown. To elucidate the function of non-canonical Arps, we focus on *Arp53D*, which is highly expressed in testes and retained throughout *Drosophila* evolution. We show that Arp53D localizes to fusomes and actin cones, two germline-specific actin structures critical for sperm maturation, via a unique N-terminal tail. Surprisingly, we find that male fertility is not impaired upon *Arp53D* loss, yet population cage experiments reveal that *Arp53D* is required for optimal fitness in *Drosophila melanogaster*. To reconcile these findings, we focus on *Arp53D* function in ovaries and embryos where it is only weakly expressed. We find that under heat stress *Arp53D*-knockout (KO) females lay embryos with reduced nuclear integrity and lower viability; these defects are further exacerbated in *Arp53D*-KO embryos. Thus, despite its relatively recent evolution and primarily testis-specific expression, non-canonical *Arp53D* is required for optimal embryonic development in *Drosophila*.

*For correspondence:
court.mschroeder@gmail.com

**Competing interests:** The authors declare that no competing interests exist.

## Introduction

Actin is an ancient, highly conserved protein that performs many cytoplasmic and nuclear functions vital for eukaryotes, including division, motility, cargo transport, DNA repair, and gene expression (*Dominguez and Holmes, 2011*; *Schrank et al., 2018*; *Wei et al., 2020*). Its origin predates eukaryotes (*Goodson and Hawse, 2002*; *Muller et al., 2005*); both bacteria and archaea encode actin-like proteins (*van den Ent et al., 2001*; *Izoré et al., 2016*). Actin forms many protein-protein interactions, including with other actin monomers, to perform its various functions (*Dominguez and Holmes, 2011*). Because of its interactions and functional importance, actin evolves under stringent evolutionary constraints (*Goodson and Hawse, 2002*; *Muller et al., 2005*). For example, despite being separated by 800 million years of evolution, actin proteins from *Drosophila melanogaster* and *Homo sapiens* are 98% identical. In addition to actin, most eukaryotes encode an expanded repertoire of actin-related proteins (Arps) because of ancient gene duplications (*Goodson and Hawse, 2002*; *Muller et al., 2005*). These Arps have specialized for a wide range of functions, including regulation of actin (Arps 2/3) (*Mullins et al., 1998*), chromatin remodeling (Arps 4–8) (*Harata et al., 2000*; *Blessing et al., 2004*; *Klages-Mundt et al., 2018*), and microtubule-based transport (Arps 1 and 10) (*Muhua et al., 1994*; *Lee et al., 2001*). Although all Arps maintain a conserved actin fold, they have specialized for their novel roles via distinct structural insertions (*Liu et al., 2013*; *Chen and Shen, 2007*). These 'canonical' Arps significantly diverged from each other early in eukaryote evolution, but now evolve under stringent evolutionary constraints, like actin.

Many eukaryotic genomes also encode evolutionarily young, rapidly evolving 'non-canonical' Arps. Unlike cytoplasmic actin and canonical Arps, which are ubiquitously expressed, non-canonical Arps appear to be exclusively expressed in the male germline (*Machesky and May, 2001*). The first described 'non-canonical' Arp was *D. melanogaster* Arp53D (named for its cytogenetic location), which was shown to be most highly expressed in the testis (*Fyrberg et al., 1994*). Its presence only in *D. melanogaster* and its unusual expression pattern led to *Arp53D* being mostly ignored in studies of cytoskeletal proteins. However, phylogenomic surveys reveal that 'non-canonical' Arps are not as rare as previously believed. Recently, we described a 14-million-year-old *Drosophila* clade that independently acquired four non-canonical *Arp* genes that are all expressed primarily in the male germline (*Schroeder et al., 2020*). Mammals also encode at least seven non-canonical Arps that are predominantly expressed in the testis (*Heid et al., 2002*; *Tanaka et al., 2003*; *Hara et al., 2008*; *Boëda et al., 2011*; *Fu et al., 2012*), at least some of which localize to actin structures in sperm development (*Hara et al., 2008*; *Boëda et al., 2011*). Thus, accumulating evidence suggests that non-canonical Arps play fundamentally distinct cytoskeletal functions from canonical Arps, which might explain both their tissue specificity as well as their unusual evolution.

To gain insight into the functions of non-canonical Arps, we performed evolutionary, genetic, and cytological analyses of *Arp53D* in *D. melanogaster*. We showed that *Arp53D* is conserved over 65 million years of *Drosophila* evolution, suggesting that it performs a critical function. Unlike actin or canonical Arps, we found that *Arp53D* has evolved under positive selection. Our cytological analyses reveal that Arp53D specifically localizes to the fusome and actin cones, two specialized actin structures found in the male germline. We show that Arp53D's unique 40 amino acid N-terminal extension (relative to actin) is necessary and sufficient to recruit it to germline actin structures. Its abundant expression in testes, together with its specialized localization, led us to hypothesize that *Arp53D* loss would lower male fertility. Contrary to this prediction, we found that *Arp53D* knockouts (KO) exhibit increased male fertility. The detrimental effect of *Arp53D* presence on male fertility is at odds with its long-term retention in *Drosophila*. Indeed, population cage experiments confirm that wildtype *Arp53D* has a net fitness benefit in populations relative to Arp53D KOs, despite the increased fertility of KO males. Seeking to explain this paradox, we investigated whether *Arp53D* also has functions outside the male germline. Despite its low expression in females and early embryos, we find that loss of *Arp53D* in the female lowers embryonic viability under heat stress. Our study finds that a non-canonical 'testis-expressed' Arp is evolutionarily retained throughout *Drosophila* for critical roles outside the male germline.

## Results

### *Arp53D* encodes a rapidly evolving non-canonical Arp that has been retained for over 65 million years

*Arp53D* was first identified as a male-specific Arp gene on chromosome 2 of *D. melanogaster* (*Fyrberg et al., 1994*). It was subsequently shown to be phylogenetically more closely related to *actin* than to any of the canonical Arps (*Goodson and Hawse, 2002*). A subsequent study proposed that *Arp53D* arose from retroduplication of *Act88F*, which encodes a *Drosophila* muscle actin (*Bai et al., 2007*). However, *Arp53D* is almost equally similar to *Act88F* (59.76%) as *Act5C* (59.2%), which encodes a cytoplasmic actin, at the nucleotide level. *Arp53D* was not found in any other non-insect genomes in a broad survey of eukaryotes, raising the possibility that it only exists in a few *Drosophila* species. To date its evolutionary origin, we investigated *Arp53D* presence in sequenced genomes from *Drosophila* and closely related *Diptera* (*Drosophila 12 Genomes Consortium et al., 2007*; *Chen et al., 2014*; *Zhou and Bachtrog, 2012*; *Renschler et al., 2019*; *Kurek et al., 1998*). Using phylogeny and shared synteny, we found clear orthologs in species as divergent as *Drosophila lebanonensis* (also known as *Scaptodrosophila lebanonensis*) but not in more divergent Dipteran species such as *Ceratitis capitata*, *Glossina morsitans*, or *Aedes aegypti* (*Figure 1A*, *Figure 1—figure supplement 1A*, *Table 1*). Thus, we found that the *Arp53D* gene arose approximately 65 million years ago at the origin of *Drosophila* (*Drosophila 12 Genomes Consortium et al., 2007*). Its retention for 65 million years implies that *Arp53D* performs an important function in *Drosophila*; deleterious or non-functional genes are quickly pseudogenized and lost within a few million years in *Drosophila* genomes (*Lynch and Conery, 2000*).

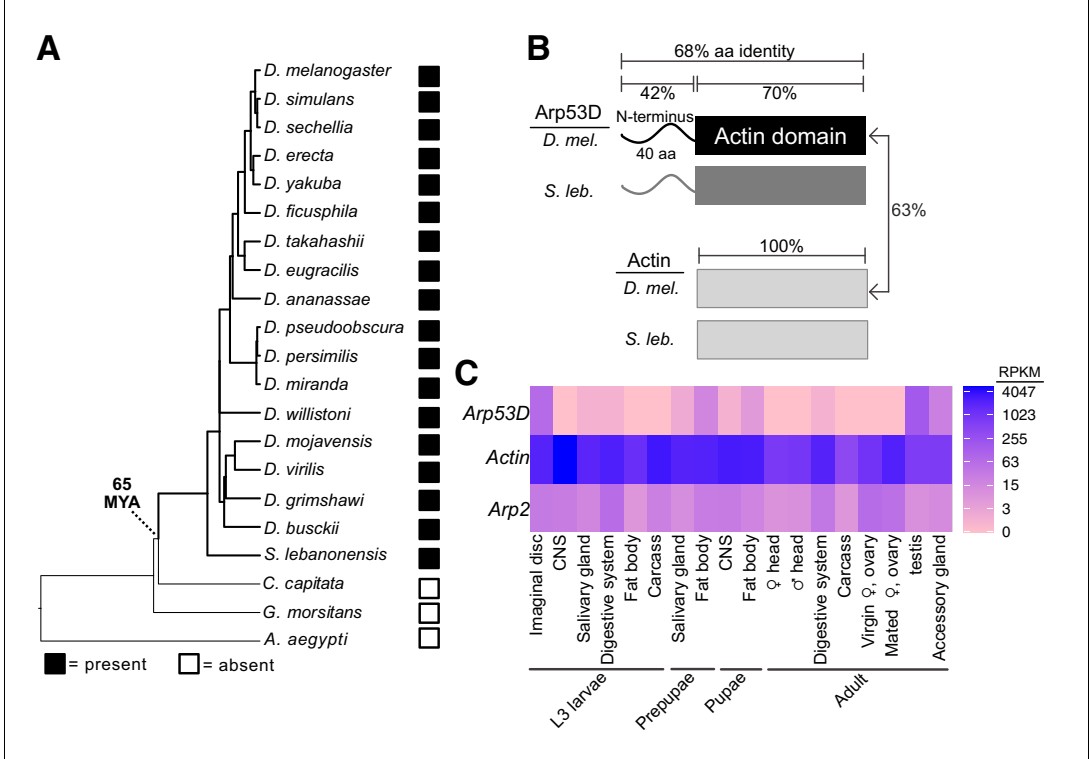

**Figure 1.** *Arp53D* encodes a rapidly evolving non-canonical Arp with male-enriched expression. (**A**) A species tree of selected *Diptera*, including 18 *Drosophila* species, *Scaptodrosophila lebanonensis*, *Ceratitis capitata*, *Glossina morsitans*, and *Aedes aegypti*, which either encode or lack *Arp53D* orthologs (filled and empty boxes, respectively). Based on this, we estimate *Arp53D* is at least 65 million years old. (**B**) Arp53D domains include an extended N-terminus, which is predicted to be unstructured, as well as a canonical actin domain. The protein identities are displayed for the different domains of actin (Act5C) and Arp53D from *D. melanogaster* and *S. lebanonensis*. Arp53D's sequence has diverged from actin and has higher between-species divergence than actin. (**C**) Expression levels from RNA-seq (in RPKM) are displayed for tissues at different developmental stages (wandering L3 larvae, white prepupae, pupae, and adults) (*modENCODE Consortium et al., 2009*; *Thurmond et al., 2019*), with blue indicating highest expression. Unlike actin and canonical Arps like *Arp2*, *Arp53D* expression is highly skewed towards males in adult flies.

The online version of this article includes the following figure supplement(s) for figure 1:

**Figure supplement 1.** *Arp53D* diverged in sequence and expression from actin.

To gain insight into its function, we compared the domain architectures of *D. melanogaster* Arp53D to cytoplasmic actin Act5C. Like canonical Arps, Arp53D includes an actin fold domain, which consists of four subdomains and a central ATP-binding pocket. However, in contrast to actin, Arp53D has an extended 40-amino acid N-terminal domain that is predicted to be mostly unstructured (*Figure 1B*, *Figure 1—figure supplement 1B*). All *Arp53D* orthologs encode this extended N-terminal domain, which is also the most rapidly evolving segment of Arp53D in sequence and length. For example, N-terminal domains from *D. melanogaster* and *S. lebanonensis* Arp53D proteins are only 42% identical, whereas the actin fold domain is 70% identical (*Figure 1B*). In contrast to Arp53D, actin homologs are 100% identical over a comparable period of evolutionary divergence. We found no homology between the N-terminal region of Arp53D to any coding or non-coding sequence in any *Drosophila* (or other) genome. The ancient evolutionary origin of *Arp53D* does not allow us to determine whether *Arp53D*'s unique N-terminus was acquired from the intergenic DNA sequence upon retroduplication or via subsequent insertions after retroduplication.

Since actin evolves under extremely strong selective constraint, the higher between-species divergence of *Arp53D* could simply reflect more relaxed selective constraints. Alternatively, it could reflect a faster than expected divergence of *Arp53D* due to diversifying selection. To distinguish between these possibilities, we took advantage of publicly available sequences of hundreds of *D. melanogaster* strains (*Lack et al., 2015*; *Lack et al., 2016*) (http://www.popfly.org, *Hervas et al., 2017*) to carry out McDonald–Kreitman (MK) tests for positive selection (*McDonald and Kreitman,*

**Table 1.** *Arp53D* orthologs used in phylogenetic analysis.

| Species | NCBI Accession or Flybase (*Thurmond et al., 2019*) gene name |
|---|---|
| *D. melanogaster* | FBgn0011743 |
| *D. simulans* | XM_016168248.1 |
| *D. sechellia* | XM_032716929.1 |
| *D. erecta* | FBgn0112814 |
| *D. yakuba* | FBgn0229606 |
| *D. eugracilis* | XM_017223499.1 |
| *D. takahashii* | XM_017160173.1 |
| *D. ficusphila* | XM_017189170.1 |
| *D. ananassae* | XM_001960587.3_modified[*] |
| *S. lebanonensis* | XM_030513294.1 |
| *D. busckii* | XM_017981926.1 |
| *D. mojavensis* | XM_002006572.3 |
| *D. virilis* | FBgn0208134 |
| *D. grimshawi* | XM_001995276.2_modified[*] |
| *D. willistoni* | FBgn0217915 |
| *D. pseudoobscura* | FBgn0078861 |
| *D. persimilis* | XM_002026570.2 |
| *D. miranda* | XM_033397705.1 |

[*]Some NCBI gene models were incomplete and were corrected using the BLAT tool (*Kent, 2002*) in UCSC's genome browser (http://genome.ucsc.edu).

*1991*). The MK test compares the ratio of non-synonymous (amino acid replacing, $P_N$) to synonymous ($P_s$) polymorphisms within a species (*D. melanogaster*) to fixed differences between species ($D_N$ and $D_S$, *D. melanogaster-D. simulans*); we exclude low-frequency polymorphisms since they have not been as strongly subject to selective scrutiny (*Fay et al., 2001*; *Bierne and Eyre-Walker, 2004*). If selective constraints are not significantly different within species versus between species, we expect $D_N:D_S$ to be approximately equal to $P_N:P_s$. If $D_N:D_S$ is greater than $P_N:P_s$, then we deduce that the gene has evolved under positive selection. Cytoplasmic actin genes have no non-synonymous changes (fixed or polymorphic); thus, they are not rapidly evolving and were not analyzed. We analyzed canonical Arps found in most eukaryotes using the MK test and found no evidence of positive selection (*Figure 1—figure supplement 1C*).

In contrast, *Arp53D* has evolved under positive selection during *D. melanogaster-D. simulans* divergence (p=0.04, *Figure 1—figure supplement 1C*), as $D_N:D_S$ (23:29) is much higher than $P_N:P_s$ (1:9). When examining the MK results in detail, we noticed that far fewer *D. melanogaster* strain sequences had passed our quality control tests for *Arp53D* than for canonical Arps. Upon further investigation, we identified a common 15 bp (five amino acid) deletion polymorphism in *Arp53D*. This deletion polymorphism could have interesting functional consequences, but also confounded our MK tests as it initially caused many strains to drop out of our analysis because their sequence contained unknown bases in this region. Redoing the MK test for *Arp53D* including all strains yielded an even more significant positive selection result (p=0.001, *Figure 1—figure supplement 1C*). The MK results indicate that at least some of the non-synonymous fixed differences between *D. melanogaster* and *D. simulans Arp53D* are adaptive substitutions, and these changes are distributed throughout the whole gene, including several in Arp53D's unique N-terminus (*Figure 1—figure supplement 1D*). These findings imply that the higher rate of Arp53D protein evolution is not simply a result of relaxed selective constraints; some of these changes have likely been evolutionarily advantageous.

These MK results revealed that *Arp53D* evolved under positive selection in recent evolutionary time (since *D. melanogaster* and *D. simulans* divergence) but do not pinpoint which residue changes

**Table 2.** RNA-seq databases analyzed.

| Species | Female | Male | Male carcass | Testis |
|---|---|---|---|---|
| D. melanogaster | SRR3123319 *Luo et al., 2020* | SRR3123321 *Luo et al., 2020* | SRR2021000 *Rogers et al., 2014* | SRR11341471 |
| D. simulans | SRR9025064 | SRR9025061 | SRR330567 | SRR9025060 |
| D. yakuba | SRR166821 | SRR6161781 *Ma et al., 2018* | SRR1693754 *Rogers et al., 2014* | SRR934057 |
| D. ananassae | SRR7243228, SRR5639307 *Yang et al., 2018*; *Benner et al., 2019*; *Mahadevaraju et al., 2021* | SRR6161785 *Ma et al., 2018* | SRR2021005 *Rogers et al., 2014* | SRR2021004 *Rogers et al., 2014* |
| D. pseudoobscura | DRR055272 *Nozawa et al., 2016* | DRR055274 *Nozawa et al., 2016* | DRR055274 *Nozawa et al., 2016* | DRR055270 *Nozawa et al., 2016* |
| D. willistoni | SRR5639517, SRR7243438 *Yang et al., 2018*; *Benner et al., 2019*; *Mahadevaraju et al., 2021* | SRR6161775 *Ma et al., 2018* | - | SRR7243415, SRR5639494 *Yang et al., 2018*; *Benner et al., 2019*; *Mahadevaraju et al., 2021* |
| D. virilis | SRR7243394, SRR5639473 *Yang et al., 2018*; *Benner et al., 2019*; *Mahadevaraju et al., 2021* | SRR6161774 *Ma et al., 2018* | SRR5278991 | SRR5278986 |
| D. mojavensis | SRR7243269, SRR5639348 *Yang et al., 2018*; *Benner et al., 2019*; *Mahadevaraju et al., 2021* | SRR6161773 *Ma et al., 2018* | - | SRR5639328, SRR7243249 *Yang et al., 2018*; *Benner et al., 2019*; *Mahadevaraju et al., 2021* |
| D. grimshawi | SRR7253580, SRR3355287 *Yang et al., 2018* | SRR7253581 *Yang et al., 2018* | - | SRR3355234, SRR7253527 *Yang et al., 2018* |
| S. lebanonensis | SRR9691967, SRR9691970 | - | SRR9691966 | SRR9691965 |

were functionally important. We wondered whether positive selection acted recurrently upon a subset of Arp53D residues over a longer period of *Drosophila* evolution. We therefore carried out maximum likelihood analyses using the PAML suite's CODEML algorithm. We found no evidence for recurrent positive selection on any *Arp53D* codons (*Figure 1—figure supplement 1E*). This suggests that the signature of positive selection does not recur in the same subset of residues. Overall, our evolutionary analyses find that *Arp53D* is an evolutionarily young, non-canonical Arp that is subject to long-term retention and atypical selective constraints, consistent with it performing a distinct function from canonical Arps.

## Arp53D localizes to specific actin structures late in sperm development

*Arp53D* was first shown to be expressed in *D. melanogaster* testes (*Fyrberg et al., 1994*). We took advantage of transcriptomic profiling of various adult tissues in *D. melanogaster* and nine other *Drosophila* species to investigate tissue-specific expression of *Arp53D*. Confirming previous analyses, we found that all *Drosophila* species show significantly male-biased expression of *Arp53D* and almost undetectable expression in adult females (*Figure 1—figure supplement 1F*, *Table 2*; *Benner et al., 2019*; *Luo et al., 2020*; *Ma et al., 2018*; *Mahadevaraju et al., 2021*; *Nozawa et al., 2016*; *Rogers et al., 2014*; *Yang et al., 2018*). In all these cases, *Arp53D* RNA expression is much higher in the testis than the remaining male carcass (*Figure 1—figure supplement 1F*, *Table 2*). More extensive transcriptome profiling of various tissues in *D. melanogaster*, obtained from the ModENCODE project (*modENCODE Consortium et al., 2009*), revealed that *Arp53D* RNA is modestly expressed in other tissues, including fat bodies and imaginal discs at earlier developmental stages (*Figure 1C*). This extremely sex- and tissue-biased expression of *Arp53D* is unusual, as cytoplasmic *actin* or canonical *Arps* are ubiquitously expressed in all tissues (*Figure 1C*).

We investigated Arp53D localization in *D. melanogaster* testes, where it is most abundantly expressed. *Drosophila* testes contain numerous cell types, including somatic cells and germ cells at many stages of development (i.e., mitotic cells, meiotic cells, and mature sperm) (*Fabian and Brill,*

*2012*). Germ cells undergo incomplete cytokinesis during their four mitotic divisions and subsequent meiosis, resulting in a cyst of 64 sperm cells, which share the same cytoplasm and membrane until full maturation (*Fabian and Brill, 2012*; *Figure 2A*). Multiple cysts at different stages of spermatogenesis are visible in the testis, allowing simultaneous visualization of all developmental stages.

We generated a transgenic fly line with superfolder GFP-tagged *Arp53D* (*sfGFP-Arp53D*; *Pédelacq et al., 2006*) under the control of its endogenous promoter (*Figure 2B*). We tagged Arp53D at the N-terminus because C-terminal tags disrupt polymerization of canonical actin (*Brault et al., 1999*). This transgene was introduced and assayed in an *Arp53D*-knockout background (described in detail later) such that only two copies of *sfGFP-Arp53D* are present, ensuring that every Arp53D molecule is fluorescently tagged. We found that sfGFP-Arp53D is undetectable during mitosis but is present within the meiotic and post-meiotic spermatocyte cysts (*Figure 2*, *Figure 2—figure supplement 1A, B*) where it localizes specifically to two germline-specific actin structures: the fusome during meiosis and spermatid elongation (*Figure 2*) and actin cones during sperm individualization (*Figure 3*).

The fusome is a membranous organelle that forms at all incomplete cytokinetic furrows following mitosis and meiosis. It is actin-coated and forms a large network that connects all developing spermatids, mediating cytoplasm exchange within the cyst (*de Cuevas and Spradling, 1998*; *Lin et al., 1994*; *Figure 2A*, *Figure 2—figure supplement 1C*). To ascertain sfGFP-Arp53D localization to the fusome, we fixed sfGFP-Arp53D-expressing testes and probed for the fusome-specific α-spectrin protein (*de Cuevas et al., 1996*). We found that sfGFP-Arp53D co-localizes with α-spectrin, confirming Arp53D localization to the fusome (*Figure 2C–E*). Arp53D localizes weakly to the fusome during meiosis (*Figure 2C*) but becomes progressively stronger post-meiosis (*Figure 2D, E*). Arp53D remains associated with the fusome even as it moves to one end of an elongating cyst. We conclude that Arp53D is targeted to the fusome specifically during meiosis with increased recruitment to the fusome during spermatid elongation. Arp53D's localization specifically to the fusome contrasts with that of actin, which is found both at the fusome and throughout the cyst (*Figure 2F*, *Figure 2—figure supplement 1C*).

During late stages of spermatogenesis, spermatids must separate and obtain their own individual membranes. In this process, known as individualization, each sperm head acquires a hollow cone of actin filaments when nuclear condensation is complete. All 64 cones in a cyst synchronously translocate along the axonemes of the sperm tails to push out excess cytoplasm ('cystic bulge') and encase each sperm in its own membrane (*Noguchi and Miller, 2003*; *Fabrizio et al., 1998*; *Figure 3A*). All 64 actin cones then undergo degradation along with the excess cytoplasmic components in a structure known as the 'waste bag' (*Noguchi and Miller, 2003*; *Fabrizio et al., 1998*; *Figure 3A*). When actin cones begin to polymerize (indicated by a gradual accumulation of filamentous actin), we find that sfGFP-Arp53D is enriched along the axoneme and slightly overlaps the base of sperm nuclei (*Figure 3B*). The puncta observed along the axoneme are usually observed with immunofluorescence but not live imaging, suggesting that this axonemal staining may be non-specific. Yet the GFP puncta at actin cones are consistently found with both fixed samples and live imaging, suggesting that it is not an immunofluorescence artifact. At this stage, sfGFP-Arp53D localization is very similar to actin at the base of the sperm head. However, when actin cones are fully formed, sfGFP-Arp53D is visible as a highly concentrated structure at the front of the actin cone, distinct from actin (*Figure 3C, E*). Subsequently, sfGFP-Arp53D remains associated with actin cones as they translocate down the microtubule-based axoneme (*Figure 3D*). Thus, sfGFP-Arp53D localizes to the leading edge of the actin cone (*Figure 3E, F*), which is composed of branched actin networks and is the site of active cytoplasm extrusion. In contrast, the rear of the actin cone is composed of parallel actin bundles (*Noguchi et al., 2008*; *Figure 3F*). Previous studies have shown that an actin-binding molecular motor—myosin VI—also localizes to the leading edge of actin cones (*Rogat and Miller, 2002*). Indeed, we find that a testis-specific myosin VI subunit (*Frank et al., 2006*) localizes proximally to Arp53D at the leading edge, though its distribution on the cone extends beyond where Arp53D is most concentrated (*Figure 3—figure supplement 1A*). Proteomic studies (*Wasbrough et al., 2010*) and our cytological analyses (*Figure 3—figure supplement 1B*) do not detect Arp53D in mature sperm. We therefore conclude that Arp53D protein must be degraded in the waste bag along with the rest of the actin cone apparatus.

Our cytological analyses reveal that Arp53D specifically localizes to two germline-specific actin structures in a dynamic manner. It first localizes to the fusome during meiosis (*Figure 2*). Once

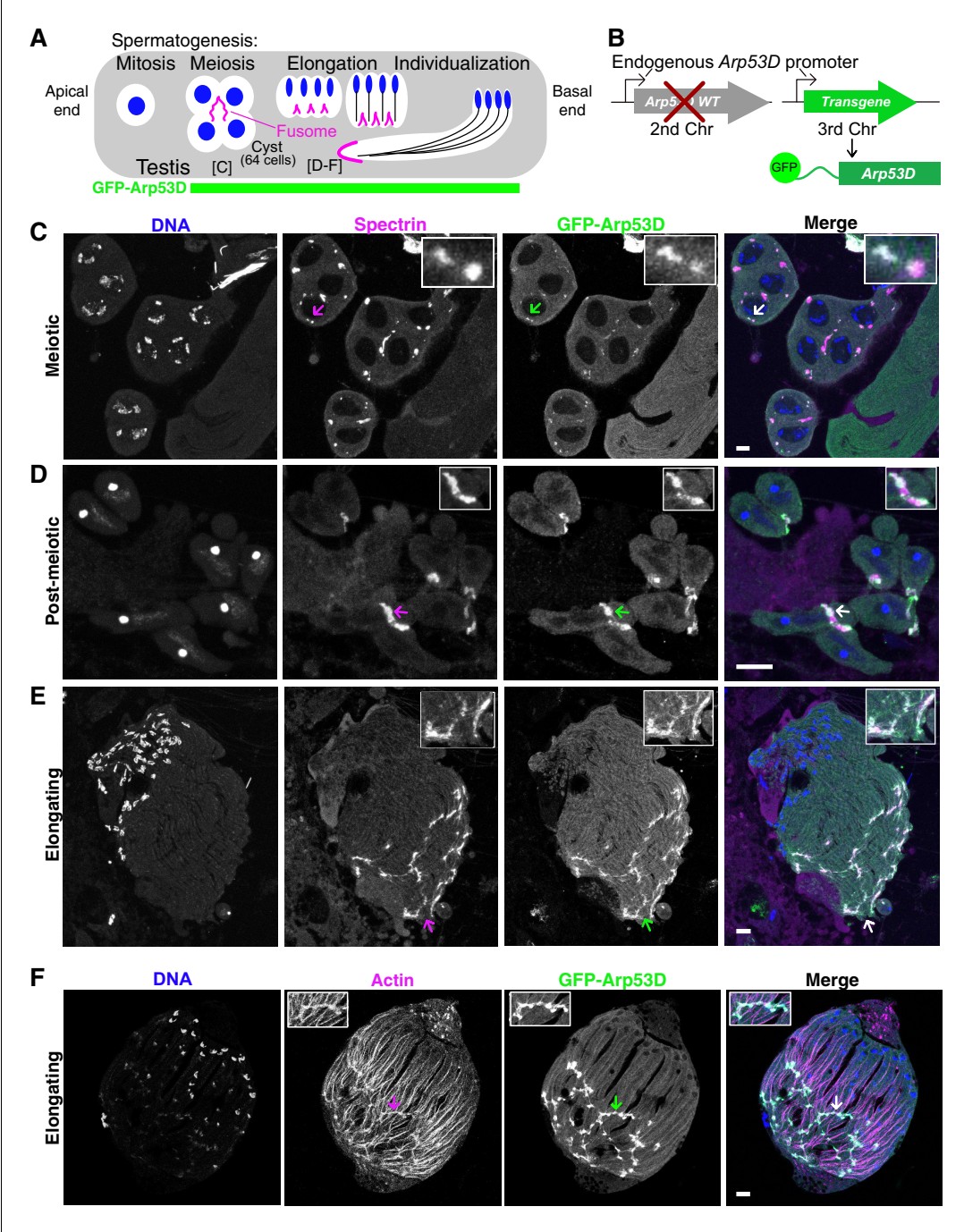

**Figure 2.** Arp53D localizes to specific actin structures late in sperm development. (A) A schematic shows spermatogenesis progression from the apical end to the basal end of the testis; green indicates stages where GFP fluorescence is visible in transgenic flies (B). Labels for the meiotic and elongating stages refer to panels (C–F). (B) To localize Arp53D in the testis, a transgene encoding Arp53D with an N-terminal superfolder GFP (sfGFP) tag was inserted on the third chromosome. The transgenic fly line was then crossed into the *Arp53D*-KO background so that transgene and knockout (KO) alleles are both homozygous; thus, all Arp53D molecules are fluorescently tagged. (C–E) Cysts from transgenic fly testes are from meiotic (C), post-meiotic (D), or elongating stages (E) of spermatogenesis. The fusome-localizing protein α-spectrin (magenta), DNA (blue), and Arp53D (green, anti-GFP) were probed. The merge of α-spectrin and Arp53D appears as white, indicating that Arp53D co-localizes with α-spectrin and thus appears at the fusome. Arrows correspond to the enlarged insets. (F) Cysts from transgenic fly testes (B) were fixed and probed for filamentous actin, indicating Arp53D co-localizes with actin only at the fusome and not throughout the cyst. All scale bars are 10 μm.

The online version of this article includes the following figure supplement(s) for figure 2:

**Figure supplement 1.** Arp53D is expressed in meiosis.

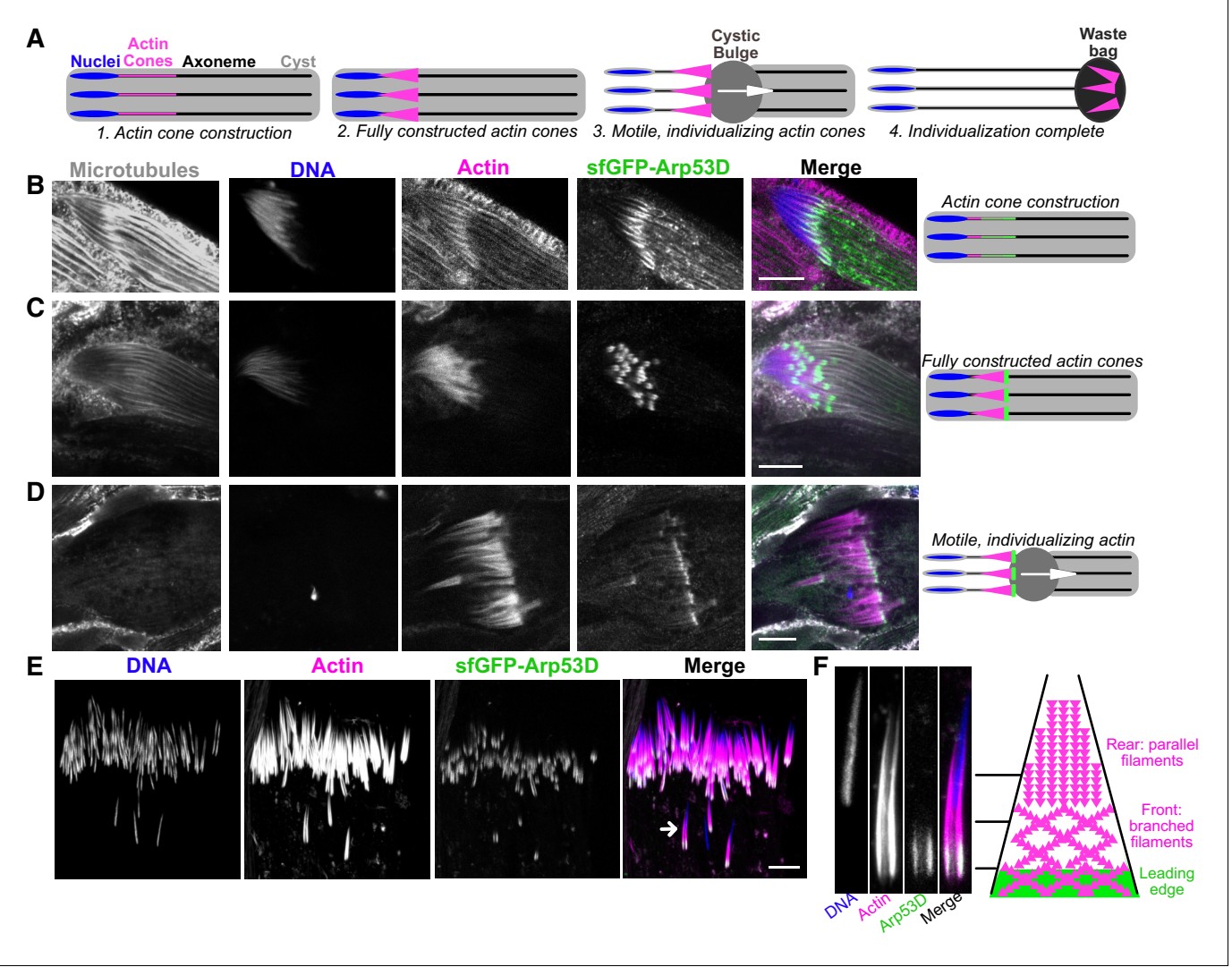

**Figure 3.** Arp53D localizes to the leading edge of actin cones during sperm individualization. (**A**) A schematic depicts the different stages of sperm individualization. Once actin cones are fully assembled at mature sperm nuclei, the cones translocate along the axoneme (a microtubule structure), pushing excess cytoplasm (the 'cystic bulge') to the end of the cyst. The cystic bulge undergoes autophagy and becomes known as the 'waste bag.' (**B–D**) Testes expressing sfGFP-Arp53D (*Figure 2B*) were dissected and fixed. Axonemal microtubules (gray, anti-tubulin), DNA (blue, DAPI), actin (magenta, phalloidin), and sfGFP-Arp53D (green, anti-GFP) were visualized. Each row shows a cyst at a different stage of individualization, which is depicted with a schematic to the right. Arp53D colocalizes with actin during cone polymerization and then coalesces at the leading edge of the cone, once the actin cone is fully constructed. Arp53D remains at the leading edge throughout translocation. (**E**) SfGFP-Arp53D-expressing testes (*Figure 2B*) were imaged live and probed for filamentous actin (SiR-actin probe; *Lukinavičius et al., 2014*) and DNA. The arrow indicates the actin cone shown in panel (**F**). (**F**) A mature sperm nucleus and its corresponding actin cone is shown in cross-section with Arp53D localizing only at the leading edge. On the right is a schematic that delineates the types of actin networks found in the cone (*Noguchi et al., 2008*) (not drawn to scale). The green filaments indicate Arp53D localization. All scale bars are 10 μm.

The online version of this article includes the following figure supplement(s) for figure 3:

**Figure supplement 1.** Arp53D localizes proximally to a testis-specific myosin VI subunit.

spermatid elongation is complete, Arp53D moves to actin cones as they are being constructed (*Figure 3*) and remains associated with actin cones until it is ultimately destroyed along with the rest of the actin cones following the completion of sperm individualization. Notably, for most of spermatogenesis, Arp53D localization is distinct from actin, which localizes more broadly. Thus, Arp53D appears to carry out specialized roles at unique cytoskeletal machineries during spermatogenesis.

## Arp53D's unique N-terminal extension is necessary and sufficient for recruitment to germline cytoskeletal structures

We investigated whether Arp53D's unique 40-residue N-terminal domain mediates its specialized localization to the fusome and actin cones (*Figure 1B*). We generated a *sfGFP-△N-term D. melanogaster* transgenic line encoding *sfGFP-Arp53D* with 35 amino acids of the N-terminal domain deleted (*Figure 4A*). The *sfGFP-△N-term* transgene was driven by the endogenous *Arp53D* promoter from the same insertion site in the fly genome as our full-length *sfGFP-Arp53D* transgene (*Figure 2B*). We dissected testes from the transgenic flies and performed immunoblotting analyses, which showed that the smaller deletion protein is expressed at comparable levels to sfGFP-Arp53D (*Figure 4—figure supplement 1B*). Moreover, like full-length *sfGFP-Arp53D* transgenic flies, *sfGFP-△N-term* transgenic flies also express GFP in meiosis (*Figure 4—figure supplement 1A*). However, unlike the full-length Arp53D fusion, localization of sfGFP-△N-term remained diffuse; we did not detect concentrated GFP signal at the fusome or actin cones (*Figure 4B, C*, *Figure 4—figure supplement 1C*). The sfGFP-△N-term protein may be less stable than full-length Arp53D, yet we believe the actin-like domain is most likely as stable as canonical actin. Based on the cytology, we conclude that the N-terminus is necessary for Arp53D's localization to these specialized germline actin structures. Since sfGFP-△N-term-Arp53D was not detected at any actin structure in the testis, we further conclude that Arp53D's actin fold domain is too divergent to co-polymerize with actin in vivo, at least within our detection limits.

We next tested whether the N-terminus is sufficient to confer Arp53D's localization to canonical actin. We generated an *sfGFP-Nt-actin D. melanogaster* transgenic line, encoding sfGFP-Arp53D N-terminal domain fused to canonical actin (Act5C) (*Figure 4A*). Like all previous transgenic constructs, we placed this chimeric protein under the control of *Arp53D*'s endogenous promoter and used the same genomic insertion location (*Figure 4A*). We found that this chimeric protein is expressed and localizes similarly to full-length sfGFP-Arp53D throughout spermatogenesis (*Figure 4B–D*), maintaining its association with the fusome during spermatid elongation and motile actin cones throughout individualization just like full-length Arp53D (*Figure 4D*). Furthermore, despite encoding an identical actin fold domain, this chimeric protein does not co-colocalize with actin throughout the developing cysts. Based on these findings, we conclude that the most prominent structural diversification of Arp53D—its N-terminal extension—is necessary and sufficient for recruitment of actin to the unique cytoskeletal machinery of the male germline.

However, Arp53D's N-terminal domain cannot confer this specialized localization onto other globular proteins. When we tested the localization of Arp53D's N-terminal domain fused to sfGFP alone, without an actin fold ('Nt-sfGFP', *Figure 4—figure supplement 1D*), we could only detect diffuse GFP expression and no concentrated signal at the fusome or actin cones (*Figure 4—figure supplement 1D*). We verified the construct was indeed expressed in the testis by conducting immunoblot analysis (*Figure 4—figure supplement 1E*). This implies that specialized localization to fusomes and actin cones requires both the Arp53D N-terminal domain as well as sequences or the tertiary structure of the actin fold domain.

## Loss of *Arp53D* does not impair male fertility

Based on its strict retention in *Drosophila* and its cytological localization to germline-specific actin structures in *D. melanogaster* testes, we predicted that *Arp53D* must play important roles in male fertility. To test this hypothesis, we created a KO of *Arp53D* using CRISPR/Cas9, introducing an early stop codon and a *DsRed* transgene under the control of an eye-specific promoter (*Figure 5—figure supplement 1A*). The *DsRed* transgene allowed us to track the KO allele by fluorescence microscopy. Based on the intensity of eye fluorescence, we could also distinguish heterozygous from homozygous KO flies, which are viable. We backcrossed the KO founder line to a wildtype strain (Oregon-R) for eight generations in order to isogenize the KO background with Oregon-R as much as possible (*Figure 5—figure supplement 1B*). Using sequencing, we confirmed the presence of *DsRed* in the *Arp53D* locus (*Figure 5—figure supplement 1C*). We also verified the lack of mutations or expression changes in *SOD2*, an essential gene located upstream of *Arp53D* (*Figure 5—figure supplement 1D*). Finally, we confirmed absence of *Arp53D* expression in the KO flies as well as absence of *Wolbachia*, a bacterium that can infect wildtype strains of *Drosophila* and confound fertility assays (*Serbus et al., 2008*; *Figure 5—figure supplement 1E, F*).

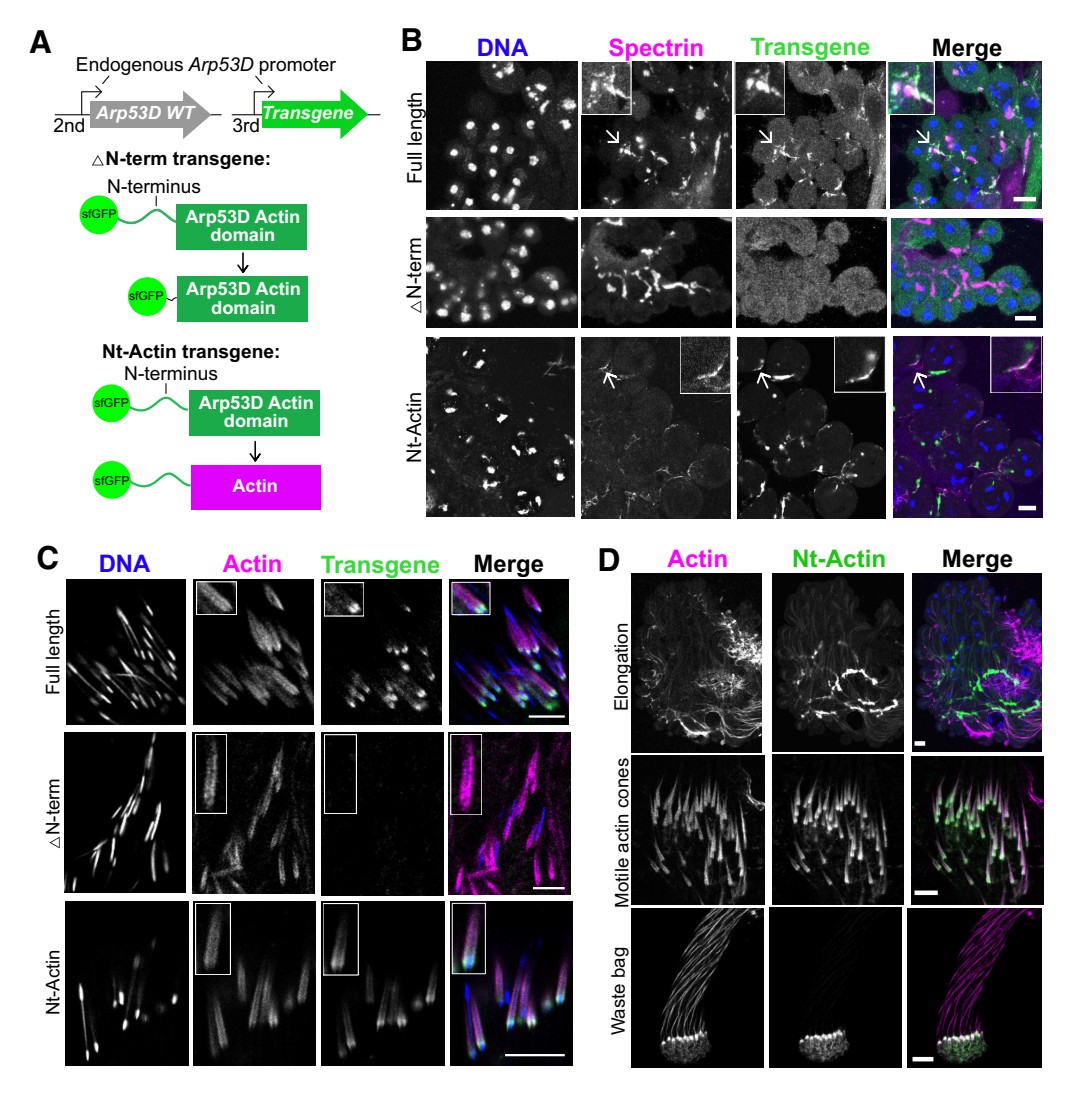

**Figure 4.** N-terminal domain of Arp53D is necessary and sufficient for localization. (**A**) Two additional transgenic fly lines were generated with the transgene on the third chromosome in the wildtype background. In the '△N-term' transgene, 35 aa of the N-terminus of Arp53D were removed and the remaining actin fold was N-terminally tagged with sfGFP. In the 'Nt-Actin' transgene, the actin domain of Arp53D was replaced with canonical actin (Act5C). Both transgenes are under the control of *Arp53D's* endogenous promoter. (**B**) Cysts from transgenic fly testes were fixed and probed with anti-GFP (green), anti-α-spectrin (magenta), and Hoechst (blue). Cysts shown are in meiotic or post-meiotic stages and indicate that Arp53D without the N-terminus can no longer localize to the fusome (α-spectrin staining), yet the Nt-Actin chimera is sufficient for localization. Arrows correspond to the enlarged insets, and all scale bars are 10 µm. (**C**) Cysts undergoing individualization were imaged live, and filamentous actin (SiR-actin probe; *Lukinavičius et al., 2014*) and DNA were labeled. Only Nt-Actin can localize to the leading edge of actin cones. (**D**) Testes from the Nt-Actin transgenic fly line were dissected and imaged live. Similar to full-length sfGFP-Arp53D, sfGFP-Nt-Actin localizes to the fusome of elongating spermatids, motile actin cones (no longer co-localizing with mature sperm nuclei), and the waste bag. All scale bars are 10 µm.

The online version of this article includes the following source data and figure supplement(s) for figure 4:

**Figure supplement 1.** Chimeric proteins reveal Arp53D's unique N-terminus is required for localization to germline actin structures.

**Figure supplement 1—source data 1.** Uncropped gel images corresponding to *Figure 4—figure supplement 1B, E*.

We reasoned that loss of Arp53D would manifest in a fertility reduction of *Arp53D*-KO males. To evaluate male fertility, we mated WT females to either homozygous *Arp53D*-KO males or isogenic WT males for 9 days and subsequently counted all progeny that survived to adulthood (all crosses are written as female × male, *Figure 5A*). This measure of male 'fertility' encapsulates number of sperm produced, their fertilization success, and successful development of sired embryos to adulthood. We were surprised to find that the KO males had significantly higher fertility than WT males at

25°C (1.3-fold increase in average progeny count, *Figure 5A*, p=0.001), which was even more pronounced at 29°C (2.4-fold increase, *Figure 5A*, p<0.0001). This increase in fertility is dose-dependent; heterozygous *Arp53D*-KO males have slightly lower fertility than KO males (*Figure 5—figure supplement 2A*). Thus, presence of only one intact copy of *Arp53D* is sufficient to reduce male fertility at 29°C (p=0.03, *Figure 5—figure supplement 2A*), while two copies are significantly worse (p<0.0001, *Figure 5A*, *Figure 5—figure supplement 2A*), suggesting that the phenotype's magnitude is dependent on *Arp53D* expression levels.

To validate our surprising findings of increased fertility in *Arp53D*-KO males, we conducted RNAi knockdown of *Arp53D* using topi-Gal4 (*Raychaudhuri et al., 2012*) to induce expression of the RNAi hairpin specifically targeted against the *Arp53D* coding region (*Figure 5—figure supplement 2B–D*). Consistent with our genetic KO findings, we found that even a partial knockdown of *Arp53D* expression resulted in significantly increased fertility at 29°C (p=0.04, *Figure 5—figure supplement 2C, D*). Together, these data reveal that lack of *Arp53D* can increase male fertility.

We hypothesized that although *Arp53D* presence intrinsically decreases male fertility, it might confer a competitive advantage in the presence of other males. To test this possibility, we mated WT females to both WT males and *Arp53D*-KO males in the same vials (*Figure 5B*). If WT and *Arp53D*-KO males had equal probabilities of successful fertilization, then 50% of adult progeny would be fathered by WT or *Arp53D-KO* males (*Figure 5B*). However, we found that *Arp53D*-KO males sired nearly 70% of the progeny in the presence of WT males, implying that they had a significant fertility advantage even in a competitive situation (p<0.0001, *Figure 5B*, *Supplementary file 1*). Our experiments show that *Arp53D* presence can be significantly deleterious to male fertility, both in isolation as well as in competition.

One possible consequence of Arp53D loss in KO males could be gross disruption of the germline actin structures to which it localizes. Contrary to this expectation, we found no gross defects in overall organization or actin intensity of actin cones (*Figure 5C*) or the fusome in *Arp53D*-KO males (*Figure 5—figure supplement 2E*). We assessed whether *Arp53D*-KO flies have increased fertility because they produce more sperm than WT flies by staining for DNA in the seminal vesicle, where mature sperm are deposited (*Figure 5—figure supplement 2F*). We did not find a significant difference between *Arp53D-KO* and WT males in seminal vesicle size, suggesting that they produced roughly equal amounts of sperm (*Figure 5—figure supplement 2G*). However, when we observed sperm development and compared the number of actin cones in WT versus KO testes, we found that the *Arp53D-KO* males had significantly more cysts with actin cones per testis than WT males (*Figure 5D*), suggesting that sperm production is accelerated upon loss of *Arp53D*.

We next sought to determine if the *Arp53D-KO*'s increase in male fertility is specific to the *Arp53D* locus, rather than being due to any off-target CRISPR mutations. When generating the KO flies, we inserted an attP site, which serves as a 'landing site' for transgenes into the *Arp53D* locus (*Figure 5—figure supplement 1A*). We took advantage of this attP site to reinsert tagless WT *Arp53D* under the control of its endogenous promoter into the *Arp53D-KO* fly line that was isogenized in the Oregon-R background (*Figure 5E*, *Figure 5—figure supplement 2H*). We found that male fertility only showed a slight decrease upon presence of the *Arp53D* rescue transgene (not statistically significant, *Figure 5F*). We attribute the lack of a significant fertility rescue to the lower expression of the rescue transgene compared to endogenous *Arp53D* in the Oregon-R background (*Figure 5—figure supplement 2I*). This apparent dependence on high expression is consistent with our previous observation that heterozygous males have fertility that is closer to KO males than to WT males (*Figure 5—figure supplement 2A*). The alternative possibility is that the male fertility effect is independent of *Arp53D*. However, this scenario would require a distinct gain-of-function mutation for male fertility in a gene that is closely linked to the *Arp53D*-KO to have survived repeated backcrossing. Although we cannot formally rule out this latter possibility due to lack of a robust effect of the *Arp53D* rescue transgene on male fertility, we find it very unlikely. In either case, we can unambiguously conclude that loss of *Arp53D* does not impair male fertility, despite testes being the primary tissue of *Arp53D* expression. Our findings thus still leave unanswered the question of why *Arp53D* was largely retained over 65 million years of *Drosophila* evolution.

## Loss of *Arp53D* results in an overall fitness disadvantage

We found that *Arp53D* loss does not reduce male fertility, yet *Arp53D* has been retained throughout most of *Drosophila* evolution, suggesting that its presence must have positive consequences. We,

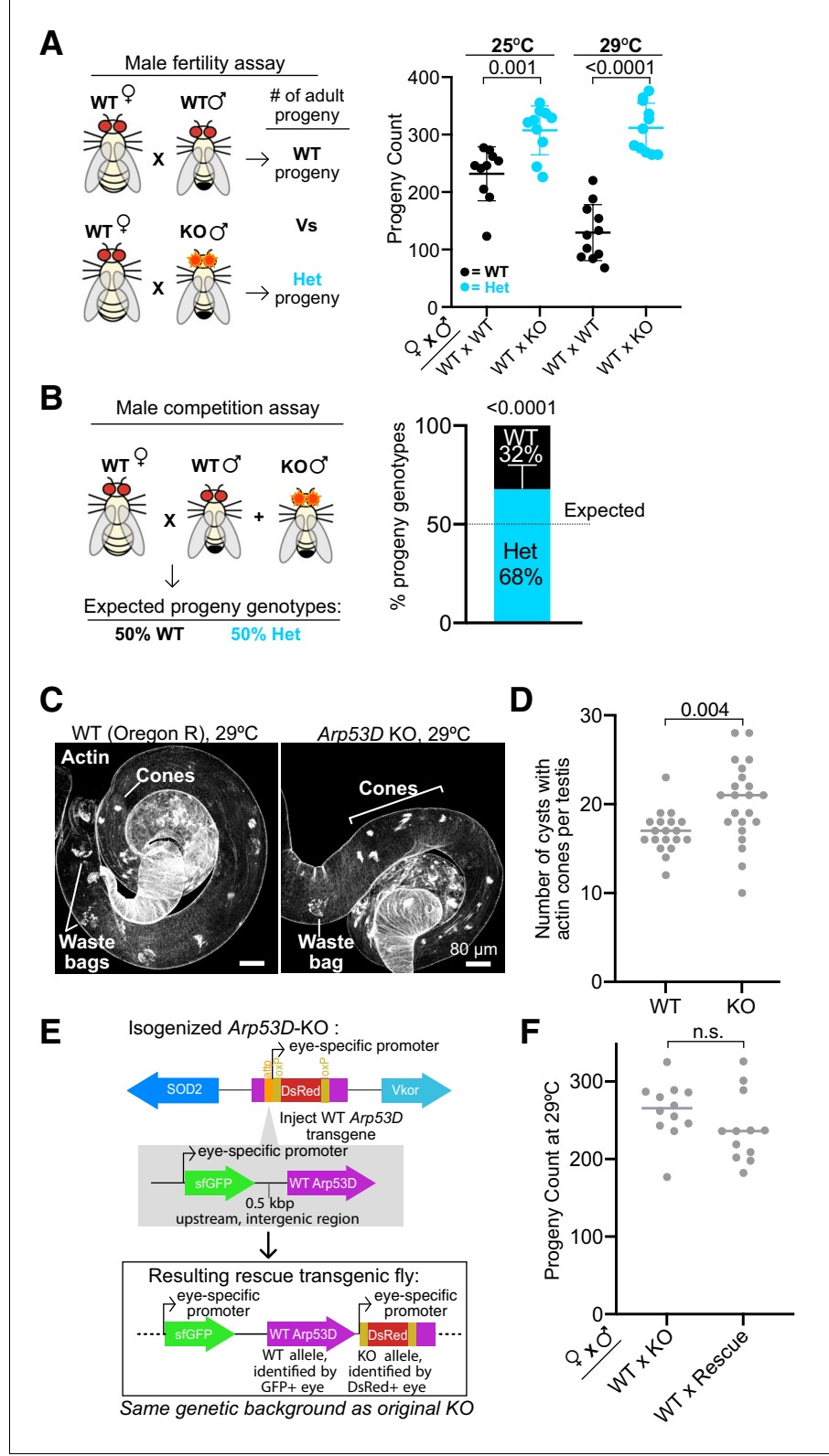

**Figure 5.** Loss of Arp53D does not impair male fertility. (**A**) Male fertility assays at 25°C and 29°C were conducted using wildtype (WT) females mated to either WT males or knockout (KO) males (all crosses are reported as female × male). All crosses were conducted in the *D. melanogaster* Oregon-R strain background, into which *Arp53D*-KO alleles were isogenized. Embryos were laid for 9 days and adult progeny were counted. KO males appeared more

*Figure 5 continued*

fertile than WT even when stressed at high temperature. For all graphs, the lines indicate the mean and standard deviation. Progeny genotypes are distinguished by color, and a t-test was used to determine all p-values that are reported. (B) For male competition assays, 10 WT females were mated to 2 WT males and 2 KO males at 25°C. Progeny of KO and WT males were identified with the presence or lack of DsRed fluorescence, respectively, and progeny genotypes are displayed as a percentage of the total population. More progeny were fathered by the KO males than the WT males. To test statistical significance, the number of progeny of each genotype was summed across the replicates and compared using a chi-squared test versus the expected 50:50 proportion (dotted line) if the competing males had equal fitness. (C) Testes from WT and *Arp53D-KO* virgin males that were aged 3 days at 29°C were dissected, fixed, and probed for actin. Cones and waste bags, which exhibit degrading actin cones, are noted. No gross differences were visible in the actin cones from *Arp53D*-KO testes. Scale bars are 80 μm. (D) The number of cysts with actin cones in each testis was quantified. More individualizing cysts were found in testes from KO compared to WT males, suggesting accelerated sperm development. (E) To test for rescue of *Arp53D*-KO phenotypes, WT *Arp53D* and 0.5 kbp of the upstream intergenic region (including its endogenous promoter) was inserted into the attP site of the *Arp53D*-KO alleles previously generated (see *Figure 5—figure supplement 1A*). The WT transgene was tracked with sfGFP under the control of an eye-specific promoter, while the KO allele was tracked with DsRed. (F) Rescue transgene-bearing male KO flies were crossed to WT Oregon-R females. Embryos were laid for 9 days, and the progeny count was compared to that from male KO flies without the transgene. The average progeny was slightly reduced but not to statistically significant levels.

The online version of this article includes the following source data and figure supplement(s) for figure 5:

**Figure supplement 1.** Characterization of isogenized *Arp53D*-KO flies verifies CRISPR-Cas9 deletion.

**Figure supplement 1—source data 1.** Uncropped gel images corresponding to *Figure 5—figure supplement 1C–E*.

**Figure supplement 1—source data 2.** Uncropped gel image corresponding to *Figure 5—figure supplement 1F*.

**Figure supplement 2.** Analysis of Arp53D's impact on male fertility.

**Figure supplement 2—source data 1.** Uncropped gel images corresponding to *Figure 5—figure supplement 2B, C*.

**Figure supplement 2—source data 2.** Uncropped gel images corresponding to *Figure 5—figure supplement 2H, I*.

---

therefore, tested whether *Arp53D* loss confers any fitness disadvantage in laboratory populations. For this, we competed KO and WT alleles of *Arp53D* over multiple generations at room temperature using a population cage experiment. This experimental design is more powerful than single-generation mating experiments as it tests for more subtle fitness differences at all lifecycle stages in males and females. In this assay, we used *Arp53D*-KO flies that were isogenized in a *w1118* genetic background (six backcrosses). We used *w1118* because it lacks eye pigmentation, making detection of DsRed fluorescence more efficient. Since *w1118* flies were used as the competing 'WT' flies, KO and WT strains are isogenic except for the absence of *Arp53D* and presence of eye-expressed *DsRed* in the KO allele at the *Arp53D* locus.

We began the experiment with three replicate populations consisting of 50 KO females, 25 KO males, and 25 WT (*w1118*) males using an excess of KO flies (75%) to put the *Arp53D*-KO allele at a starting advantage (*Figure 6A*). At each generation, we randomly selected 50 females and 50 males to act as founders for the next generation (without scoring the fluorescent eye marker for the *Arp53D*-KO allele) and quantified the remaining progeny for the presence of the *Arp53D-KO* allele (*Figure 6B*). If there were no advantages associated with the WT *Arp53D* genotype, then homozygous WT flies (lacking *DsRed*) should dramatically decrease within 20 generations. In contrast to this expectation, we found a robust and consistent increase in frequency of the homozygous WT genotype across all three replicate populations despite stochastic effects due to genetic drift given the small effective population sizes. The frequency of the homozygous WT genotype reached an average proportion of 67% among the three replicate populations in just 20 generations (*Figure 6B*). This rise in frequency suggests a strong fitness disadvantage for the KO genotype. To infer the selective coefficients associated with this increased fitness, we modeled three different scenarios that varied the relative fitness of heterozygote genotypes (i.e., equal to homozygous WT, equal to homozygous KO, or intermediate between HOM and WT; see Materials and methods). Based on these models, we find that the rapid, consistent rise of the WT allele in just 20 generations is most consistent with

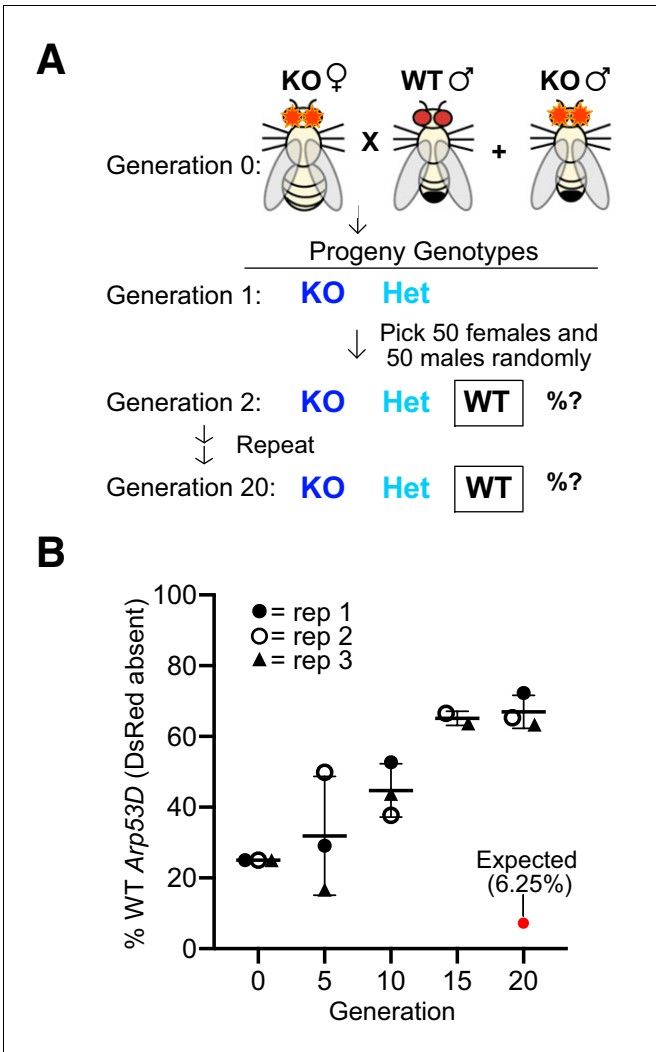

**Figure 6.** Loss of *Arp53D* results in an overall fitness disadvantage. (**A**) *Arp53D*-KOs were isogenized in the *w1118* background. A population cage experiment was initiated by mixing 50 *Arp53D*-KO females with 25 *Arp53D*-KO males, and 25 wildtype (WT) (*w1118*) males in each of three replicate bottles. All subsequent generations were passaged by randomly selecting 50 male and 50 female progeny from the previous generation and placing them in a new bottle at room temperature. (**B**) All progeny at selected generations were assessed for the presence of DsRed-fluorescent eyes, the marker for the *Arp53D*-KO allele. The graph displays the percent of each generation's total population that were homozygous for the WT allele (entirely lacking DsRed fluorescence); replicates are distinguished by icons with different shapes and color. The red dot indicates the expected percentage of homozygous WT progeny over time if no fitness advantage is associated with WT *Arp53D* (according to Hardy–Weinberg equilibrium). In contrast to this expectation, homozygous WT *Arp53D* flies overtook the majority of the population in all three replicate populations, demonstrating *Arp53D* confers a strong fitness advantage to *D. melanogaster*. WT: wildtype.

The online version of this article includes the following figure supplement(s) for figure 6:

**Figure supplement 1.** Modeling of the population cage experiment estimates the fitness disadvantage upon loss of *Arp53D*.

the first scenario, in which the frequency of the heterozygotes is the same as homozygous WT genotypes (*Figure 6—figure supplement 1*). Moreover, we infer the WT allele of *Arp53D* confers between a 30–40% selective advantage over the KO allele per generation (*Figure 6—figure supplement 1*). Thus, although *Arp53D* appears dispensable for male fertility, it must play important roles beyond the male germline in *D. melanogaster*.

## Lack of *Arp53D* reduces female fertility under heat stress

Given that *Arp53D* presence is unnecessary or even disadvantageous for male fertility, we considered whether other life history traits require *Arp53D*, which might help explain its long-term evolutionary retention. Although *Arp53D* is most abundantly expressed in adult testes, there is also weak expression in other tissues and developmental stages (*Figure 1C*). Moreover, although published transcriptomic data suggests that *Arp53D* is undetectable in adult females (*modENCODE Consortium et al., 2009*; *Figure 1C*), bulk RNA-seq analyses can miss transcripts that are expressed at low levels. We, therefore, carried out sensitive RT-PCR analyses (high number of amplification cycles), which revealed that *Arp53D* is indeed expressed in adult females, albeit at much lower levels than in males (*Figure 7—figure supplement 1A*); expression is highest in ovaries and undetectable in somatic tissues (*Figure 7—figure supplement 1B*). However, *Arp53D* expression in the ovary is much lower than in the testis (*Figure 7—figure supplement 1B*), which agrees with previous RNA-seq data that indicates very low to undetectable levels of *Arp53D* expression in the ovary (*Jevitt et al., 2020*; *Slaidina et al., 2020*). Consistent with this low expression, our cytological examination of ovaries in female flies expressing *sfGFP-Arp53D* did not reveal GFP expression above background levels (*Figure 7—figure supplement 1C*).

To investigate whether this weak expression in ovaries has important biological consequences, we crossed *Arp53D*-KO females to either WT males or *Arp53D*-KO males, and compared the number of adult progeny produced relative to WT × WT crosses (*Figure 7A*). At room temperature (25° C), we did not observe any significant differences between these three crosses (*Figure 7A*). However, at 29°C, KO × WT crosses produced significantly fewer adult progeny than WT × WT crosses (1.7-fold decrease in average progeny count, p=0.0007, *Figure 7A*). *Arp53D*-KO females therefore have a fertility disadvantage at higher temperatures, suggesting a maternal effect.

To further test for a maternal effect, we conducted two additional crosses—HET × KO and KO × HET—at 29°C (*Figure 7B*). In both crosses, the progeny genotypes produced are the same (heterozygous and homozygous KOs), whereas the parental genotypes are swapped. If there were no maternal effect, or if paternal and maternal contributions of *Arp53D* were identical, we would expect both crosses to yield the same number of progeny. In contrast to this expectation, we found that the KO female cross produced far fewer total progeny than the HET female cross (p<0.0001, *Figure 7B*). This confirms that *Arp53D* KOs exhibit a maternal effect, indicating that *Arp53D* surprisingly plays a significant role despite its weak expression in the female.

We next tested whether this reduction in fertility under heat stress could be solely attributed to loss of *Arp53D*. For this, we again used the KO 'rescue' fly line with untagged WT *Arp53D* reinserted into the *Arp53D*-KO locus (*Figure 5E*, *Figure 7C*). We found that KO females expressing the *Arp53D* rescue transgene in one or two copies had robustly increased fertility compared to KO females without the transgene (*Figure 7C*, *Figure 7—figure supplement 1D*) despite low expression of the rescue transgene (*Figure 5—figure supplement 2I*). These findings confirm that *Arp53D*'s contribution to fitness is largely driven by its maternal effect. Moreover, they confirm our predictions from the population-genetic modeling that heterozygous *Arp53D* can at least partially restore fitness.

## Lack of zygotic *Arp53D* can lead to lower fitness, which is nearly masked by maternal contributions

A strong maternal effect explains most but not all of the defects seen in crosses involving *Arp53D*-KO flies. The number of adult progeny produced at 29°C in KO × KO crosses is further reduced 3.5-fold relative to KO × WT crosses (p<0.0001, *Figure 7A*). This reduction is especially surprising since KO males have increased fertility over WT males at this temperature (*Figure 5A*). These findings suggest that the complete loss of *Arp53D* resulting from a KO × KO cross must lead to an additional zygotic disadvantage since KO × WT crosses produce heterozygous zygotes (*Figure 7A*).

To further confirm this zygotic effect, we quantified the ratio of KO versus HET progeny produced in the previous HET × KO and KO × HET crosses (*Figure 7B*). If there were no contribution of zygotic genotype to survival, we would expect a 1:1 mix of KO and HET genotypes among surviving offspring (*Figure 7B*, schematic). In contrast, we find that KO progeny only made up <30% of total progeny in the KO × HET cross (p<0.0001, *Figure 8A*, *Supplementary file 1*). Thus, KO zygotes are at a survival disadvantage relative to HET zygotes. In the reciprocal cross with HET mothers, KO

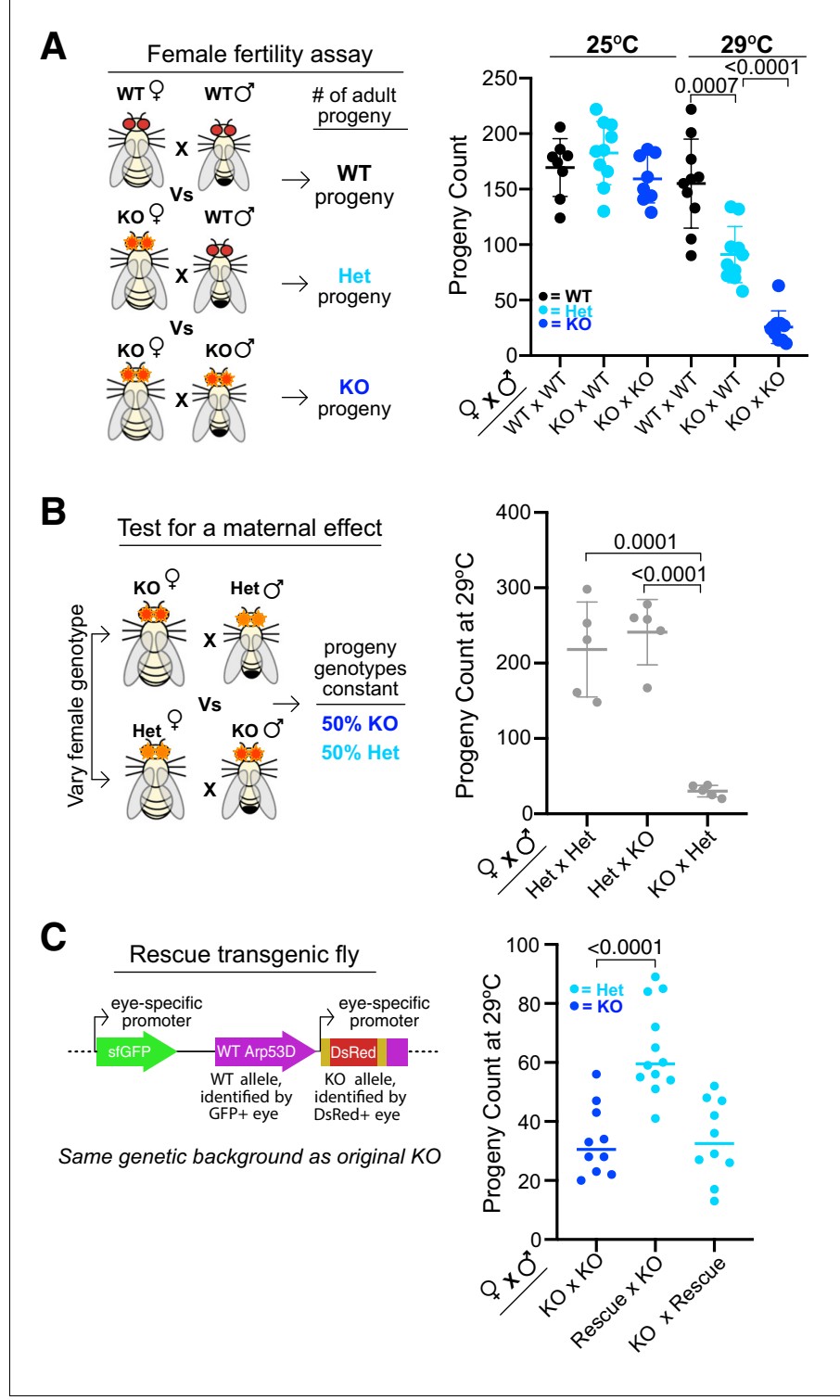

**Figure 7.** Maternal contribution of *Arp53D* is required for optimal fitness. (**A**) Female fertility assays at 25°C and 29°C were conducted with knockout (KO) females mated to either wildtype (WT) males or KO males. Matings took place for 9 days, and all resulting adult progeny were counted. KO females have fewer progeny at high temperature, especially when a KO male is present. A t-test was used to determine all p-values. (**B**) To test for a maternal effect, *Arp53D*-KO females were crossed to heterozygous (HET) males, whereas HET females were crossed to KO males in reciprocal crosses, which yield progeny with the same genotypes. The total adult progeny counts for each cross are shown. Crosses between KO females and HET males exhibit a considerably lower

*Figure 7 continued on next page*

*Figure 7 continued*

progeny count compared to the reciprocal cross between HET females and KO males (<0.0001), suggesting that the KO females exhibit a maternal effect. (**C**) KO males were crossed to either KO females or KO females encoding the homozygous *Arp53D* rescue transgene ('Rescue'), which was identified via GFP-positive eyes (see schematic). In addition, rescue transgene-bearing male KO flies were crossed to KO females. Matings took place for 6 days, and all resulting adult progeny were counted. KO females expressing a rescue transgene had more progeny than KO females, indicating a rescue of the fertility phenotype. However, KO rescue males did not exhibit an increase in progeny count when crossed to KO females, suggesting that *Arp53D* is predominantly playing a maternal role.

The online version of this article includes the following source data and figure supplement(s) for figure 7:

**Figure supplement 1.** *Arp53D* plays maternal roles.

**Figure supplement 1—source data 1.** Uncropped gel images corresponding to *Figure 7—figure supplement 1A*.

**Figure supplement 1—source data 2.** Uncropped gel images corresponding to *Figure 7—figure supplement 1B*.

---

progeny were also recovered at lower than 50% frequency (p=0.005, *Figure 8A*, *Supplementary file 1*, note that total progeny counts are 10-fold higher for this cross). Although this zygotic effect is subtler with HET mothers rather than KO mothers, it is highly consistent across replicates and significant (p=0.005, *Supplementary file 1*). Thus, loss of Arp53D in the zygote reduces survival, yet this zygotic effect can be almost entirely masked in the presence of maternal contributions of Arp53D.

We further tested the dependence of *Arp53D*'s zygotic effect on maternal *Arp53D* by conducting a separate cross between HET males and HET females (*Figure 8B*). In this scenario, all progeny receive the same *Arp53D* contribution from their HET mothers. If there is a zygotic effect that is independent of the maternal genotype, we expect that KO progeny should comprise less than a quarter of the total progeny (*Figure 8B*). However, we find that the fraction of KO progeny is almost exactly 25%, comparable to the proportion of WT progeny (*Figure 8B*, *Supplementary file 1*). Thus, *Arp53D*-dependent zygotic effects are nearly masked in the presence of maternal *Arp53D* contributions. Overall, our genetic experiments allow us to conclude that maternal contributions of Arp53D are primarily responsible for its contribution to *Drosophila* fitness. Lack of maternal contribution can only be partially rescued by zygotic expression from the paternal *Arp53D* allele (*Figures 7A* and *8C*). Loss of both maternal and zygotic *Arp53D* leads to the most significant fitness costs (*Figures 7A* and *8C*).

We investigated early embryonic expression of *Arp53D* to explain its zygotic effect. Publicly available in situ data revealed weak signal for *Arp53D* first in stages 1–3 of embryogenesis (*Figure 8—figure supplement 1*; *Jambor et al., 2015*), which precedes zygotic transcription (*Tadros and Lipshitz, 2009*) and is therefore likely the result of maternal contribution. *Arp53D* RNA is much more evident during embryonic stages 10–17, after zygotic transcription has initiated (*Figure 8—figure supplement 1*; *Jambor et al., 2015*). Single-embryo RNA-seq analyses that use single-nucleotide polymorphisms (SNPs) to distinguish between maternal and zygotic transcripts also reveal zygotic expression of *Arp53D* (*Lott et al., 2011*). Based on these data, we conclude that *Arp53D* is sufficiently expressed in embryos to manifest its zygotic effects.

## Loss of *Arp53D* impairs early embryonic development

To understand why fewer adult progeny are recovered when maternal and/or zygotic *Arp53D* is absent (*Figure 7A, B*), we compared the number of embryos laid versus the number that actually develop in WT × WT, KO × WT, and KO × KO crosses at 29°C (*Figure 9A*). We saw no significant differences in the number of eggs laid or the percent of fertilized eggs between these crosses (*Figure 9—figure supplement 1A, B*). We therefore conclude that maternal and zygotic Arp53D must be required post-fertilization and after embryos have been laid. Indeed, we found that 25% of eggs failed to develop in the KO × WT cross, whereas <20% of eggs failed to develop to larval stages in WT × WT crosses (p=0.04, *Figure 9A*). An even more dramatic effect was observed in the KO × KO crosses, in which nearly 40% of eggs failed to develop (p=0.02, *Figure 9A*). Based on these results, we conclude that *Arp53D* is required for optimal embryonic development.

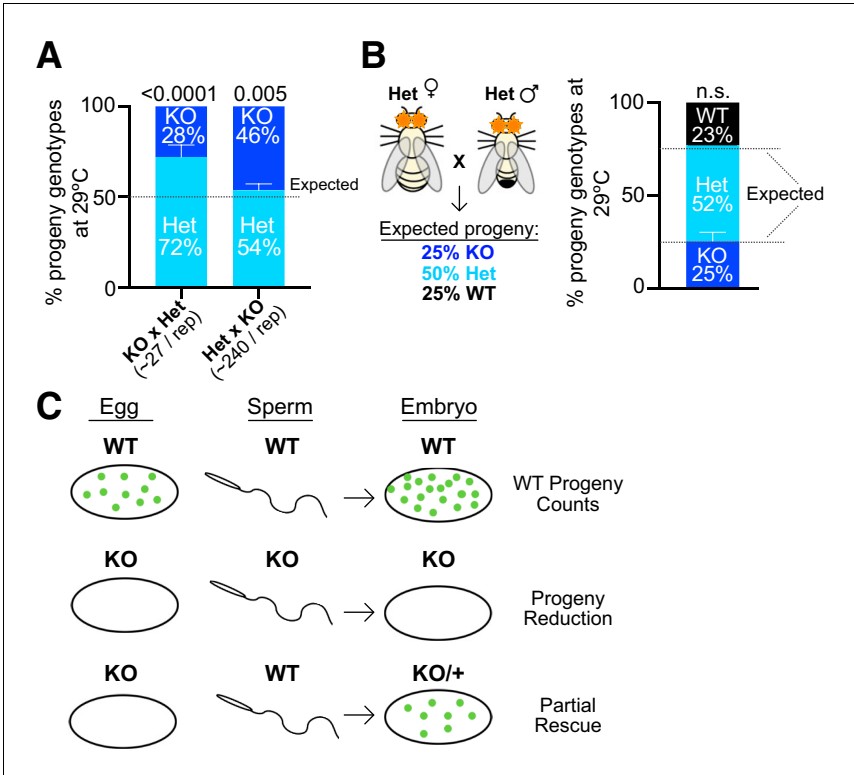

**Figure 8.** A zygotic fitness effect of *Arp53D* is masked by maternal contributions. (**A**) To assess *Arp53D*'s zygotic requirements for fitness, we quantified progeny produced from the reciprocal crosses in *Figure 7B* as a percentage of the total population. Homozygous knockout (KO) progeny were reliably distinguished from heterozygous (HET) progeny by intensity of DsRed fluorescence. Progeny fractions obtained were compared to 50:50 Mendelian expectation using a chi-squared test, and p-values are reported. KO progeny comprise a significantly lower proportion than the expected 50% of the population. (**B**) To determine if KO progeny are only at a disadvantage when the mother lacks *Arp53D*, HET females were crossed to HET males, and progeny genotypes were quantified and compared to Mendelian expectations of 25:50:25. KO progeny were present at nearly 25% of the population, indicating that KO progeny have no significant fitness disadvantage when the mother has one copy of *Arp53D* (p-values from a chi-squared test indicate deviation from Mendelian expectation and are not significant). (**C**) A model for Arp53D's role in fitness (as assessed by adult progeny counts) under heat stress. For optimal fitness, *Arp53D* must be contributed maternally or via zygotic transcription of the paternal *Arp53D* wildtype (WT) allele. Maternal contribution is most critical, while zygotic transcription alone only leads to partial rescue.

The online version of this article includes the following figure supplement(s) for figure 8:

**Figure supplement 1.** *Arp53D* is expressed in embryos.

To investigate why some KO embryos fail to develop, we allowed WT × WT and KO × KO flies to lay for 2 hr at 29°C. We fixed and stained resulting WT and KO embryos for DNA to stage embryos and identify any gross morphological defects. We found a higher incidence of abnormal nuclei that appeared disorganized and lacked compaction in *Arp53D*-KO embryos (28% in KO vs. 3% in WT, p<0.0001, *Figure 9B, C*). Following fertilization, WT *Drosophila* embryos undergo rapid mitotic divisions. Consequently, mitotic fidelity is often sacrificed, leading to damaged nuclei that are allowed to cycle but are removed from the cell cortex and deposited in the embryo's yolk at a discrete stage of embryogenesis (*Foe and Alberts, 1983*; *Sullivan et al., 1993*). This leads to gaps in an otherwise ordered array of nuclei on the surface of *Drosophila* embryos. This phenotype has been referred to as 'nuclear fallout,' which increases due to mitotic errors preceding and during late cortical nuclear cycles in *Drosophila* embryogenesis (cycles 11–14) (*Sullivan et al., 1993*). We found that *Arp53D*-KO embryos exhibited more gaps larger than 25 μm$^2$ in the cortex than WT embryos at 29°C, suggesting an increase in the occurrence and removal of damaged nuclei (*Figure 9D, E*). Moreover, *Arp53D*-KO

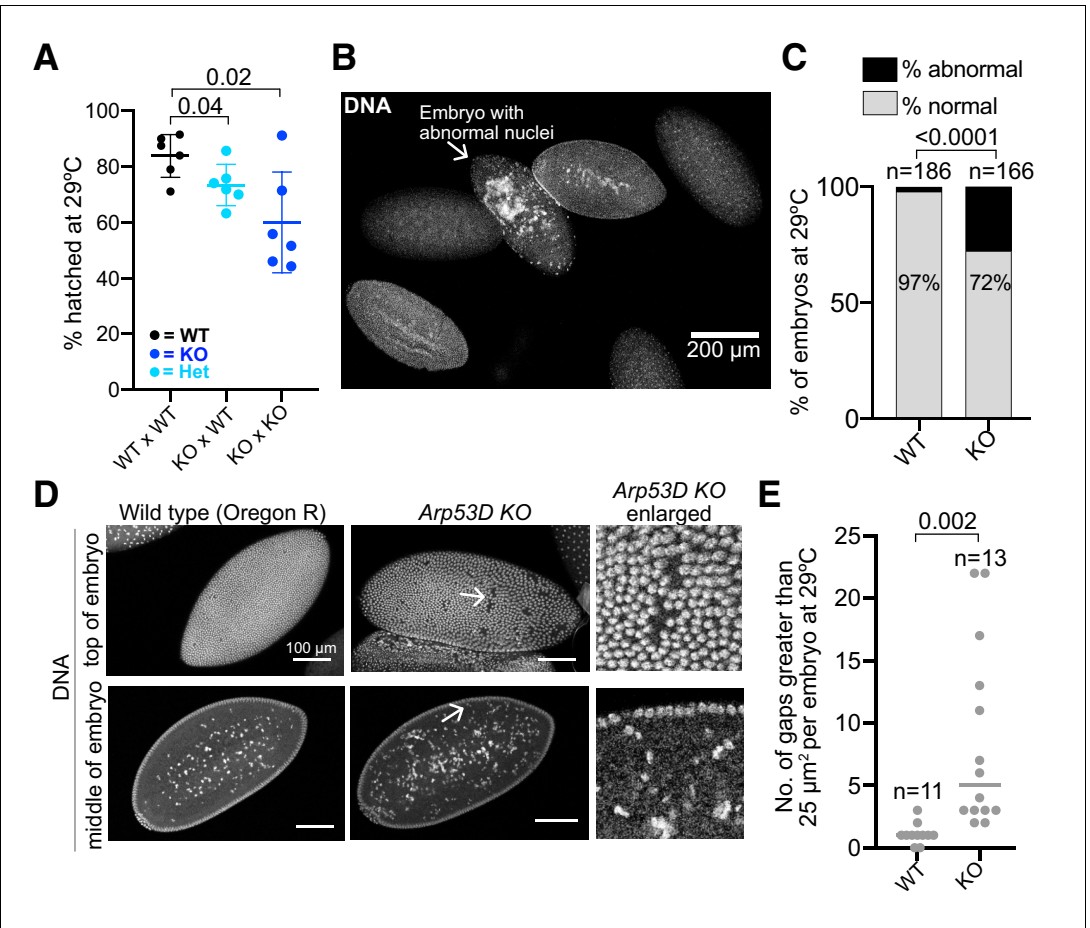

**Figure 9.** Loss of *Arp53D* impairs early embryonic development. (**A**) Knockout (KO) females were crossed to wildtype (WT) or KO males and allowed to lay for 2 hr at 29°C, and all resulting embryos were counted. Hatched embryos were then quantified 24 hr later and are displayed as a percentage of total embryos laid the previous day. KO females lead to reduced embryonic viability relative to WT females (p=0.04, 0.02). (**B, C**) After a 2 hr lay at 29°C, embryos were collected from WT Oregon-R, *Arp53D*-KOs, and *Arp53D*-KOs encoding the rescue *Arp53D* transgene. Embryos were fixed, stained for DNA, and assessed for abnormal nuclei. In the representative image of WT Oregon-R embryos, only one (arrow) exhibits disorganized and aggregated nuclei. However, *Arp53D*-KO embryos exhibited more abnormal nuclei than WT, which correlates with their reduced viability. The number of embryos quantified is denoted above each genotype. (**D**) The embryos in (**B, C**) that were at approximately cycles 13–14 of embryogenesis (*Kotadia et al., 2010*) were assessed for 'nuclear fallout,' which results in visibly large gaps in the embryo's epithelium and an increase in damaged nuclei in the middle of the embryo. The enlarged images in the third column ('Arp53D KO enlarged') correspond to the arrows in the Arp53D KO images in the second column. (**E**) Gaps were measured in WT and *Arp53D-KO* embryo epithelia represented in (**D**). The number of gaps larger than 25 µm$^2$ is significantly higher in KO embryos at 29°C.

The online version of this article includes the following figure supplement(s) for figure 9:

**Figure supplement 1.** Arp53D plays roles in embryonic development and not fertilization.

embryos have more nuclei that are presumed to be damaged in the yolk than WT embryos (*Figure 9D*). However, in 'rescued' KO embryos (bearing the *Arp53D* rescue transgene, *Figure 5E*), we find a significant reduction in the frequency of abnormal nuclei (15% in the rescue vs. 28% in KOs, p=0.005, *Figure 9—figure supplement 1C*) and a slightly lower number of large gaps per embryo (not significant, *Figure 9—figure supplement 1D*). Cytoskeletal proteins are often important in the organization and migration of nuclei during early embryogenesis (*Sullivan et al., 1993*). Our cytological analyses of embryos (*Figure 9B–E*) together with our genetic analyses (*Figures 7* and *8*) indicate that Arp53D plays a key maternal and zygotic role in embryonic development, despite being primarily testis-enriched in expression.

**Table 3.** Primers used in this study.

| Purpose | Primer 1's sequence | Primer 2's sequence |
| --- | --- | --- |
| Sequencing *Arp53-KO* locus | ACCTTCCCGAATCAAAATCGA | TTCACGTACACCTTGGAGCC |
| Sequencing WT *Arp53D* locus | AGATACTCCCCGTGCTGTCT | GCAAATCCATTGGATCCGCC |
| Testing presence of *Wolbachia* (**Schneider et al., 2014**) | TTCGCCAATCTGCAGATTAAA | GTTTTAAACGCTTGACAA |
| Sequencing *SOD2* | CTTCAGATCATCGCTGGGCT | TGAAGAATGTTCTGTGCCCGT |
| RT-PCR of *SOD2* | TGGAGCTGCATCACCAGAAG | TCTTGTTGGGCGAGAGGTTC |
| RT-PCR of *Arp53D* (Fig. S4E) | ACCTTCCCGAATCAAAATCGA | GCGGCGTGGTGTGAATTAC |
| RT-PCR of Arp53D (Supp. Fig. S7B) | CAAAATCGATATAACAAATAAAC GGGCACAGAACATTCTTCAC | GATACTTTAGGGGTTAGTATT CCCCTTTTTCGGGC |

## Discussion

Actin and canonical Arps represent some of the most conserved proteins in eukaryotic genomes. Canonical Arps diversified early in evolution and have been mostly retained for their essential cellular functions since. In contrast to these ancient, conserved Arps, many genomes also encode non-canonical Arps that are often evolutionarily young, rapidly evolving, and predominantly expressed in the male germline. These non-canonical Arps have received much less scientific attention than canonical Arps. In this study, we investigated one of the first-described non-canonical Arps, encoded by *Arp53D* in *D. melanogaster*. Although this Arp is not widely conserved even in animal genomes, we show that it has been retained through 65 million years of *Drosophila* evolution and is important for optimal fitness in *D. melanogaster*. Moreover, even though *Arp53D* is predominantly expressed in *Drosophila* testes, we find that it exerts its critical function during embryogenesis. Non-canonical Arps like *Arp53D* are found in many animal genomes, including mammals. Our analyses suggest that these non-canonical Arps might encode many important functions that have been previously overlooked.

The evolutionary invention of Arps allows the deployment of the actin fold to perform new functions without compromising actin's many essential pre-existing functions within a cell. Compared to canonical Arps, the more recent evolutionary divergence of non-canonical Arps provides a better opportunity to dissect how they diverged from actin to acquire and consolidate their varied cellular functions. For example, Arp53D is distinguished from canonical actin by its divergent actin fold domain and a longer 40 amino acid residue N-terminal domain. Although N-terminal tails in actin proteins are typically much shorter—only three amino acid residues in length—they regulate the binding of many regulatory proteins, such as myosin (**Sutoh et al., 1991**; **Hansen et al., 2000**). Moreover, post-translational modifications of the N-terminal domains can affect actin localization, polymerization, and interactions with actin-binding proteins (**Varland et al., 2019**). Our analyses show that the unique N-terminal tail is necessary and sufficient to explain Arp53D's specialization to germline-specific actin structures during spermatogenesis (**Figure 4**). We hypothesize that the longer N-terminal tail of Arp53D may allow it the ability to interact specifically with other cytoskeletal proteins, thereby distinguishing it from canonical actin.

Many non-canonical Arps show testes-enriched patterns of expression (**Schroeder et al., 2020**; **Heid et al., 2002**; **Hara et al., 2008**; **Fu et al., 2012**; **Harata et al., 2001**). It is not unexpected that novel Arps might specialize for spermatogenesis, which requires several novel cytoskeletal functions and complex actin structures. For example, *Drosophila* exhibits a unique sperm developmental program that deploys two germline-specific actin structures: the fusome and actin cones. Arp53D localizes to both in a developmental stage-specific manner that is distinct from actin. Actin cones are unique to *Drosophila* males flies, whereas the fusome is found in both *Drosophila* females and males, in additional insects (**de Cuevas et al., 1997**), and in frogs (**Kloc et al., 2004**). Although these actin structures are absent in many species' germ cell developmental programs, the actin-based processes of cytoplasm sharing and sperm separation span many phyla (**de Cuevas et al., 1997**; **Geyer et al., 2009**). The specialized requirement of these actin processes may have led to the independent origin and retention of many non-canonical Arps throughout animal evolution. Indeed, we find that another non-canonical Arp, which independently arose via gene duplication from canonical *Arp2* in the *D. pseudoobscura* lineage, also specialized to localize to actin cones (**Schroeder et al., 2020**). Their

**Table 4.** Imaging reagents.

| Antibody or chemical | Company | Purpose | Dilution |
|---|---|---|---|
| Anti-GFP (chicken) | Abcam (13970), RRID:AB_300798 | Western blot | 1:2000 |
| | | Immunofluorescence | 1:500 |
| Anti-tubulin (rabbit) | Abcam (6046), RRID:AB_2210370 | Western blot | 1:500 |
| | | Immunofluorescence | 1:200 |
| Anti-α-spectrin | Developmental Studies Hybridoma Bank (AB_528473), RRID:AB_528473 | Immunofluorescence | 1:50 |
| Anti-phospho-Histone H3 (Ser10) | Millipore (Upstate Brand), RRID:AB_310177 | Immunofluorescence | 1:1000 |
| Anti-calmodulin (rabbit) | Gift from Kathleen Beckingham and Leslie Vosshall | Immunofluorescence | 1:50 |
| Anti-mouse Cy3 or Cy5 | Invitrogen | Immunofluorescence | 1:2000 |
| Anti-rabbit Cy3 or Cy5 | Invitrogen | Immunofluorescence | 1:2000 |
| Anti-chicken 488 | Invitrogen | Immunofluorescence | 1:2000 |
| Anti-chicken 680 | LI-COR, RRID:AB_1850018 | Western blot | 1:2500 |
| Anti-rabbit 800 | LI-COR | Western blot | 1:2500 |
| Phalloidin Cy3 | Thermo Fisher | Immunofluorescence | 1:40 |
| Phalloidin Cy5 | Thermo Fisher | Immunofluorescence | 1:40 |
| SiR-actin (*Lukinavičius et al., 2014*) | Cytoskeleton, Inc | Live imaging | 10 µM |

role in reproduction may have also led to their accelerated rate of evolution due to strong selective pressures from sperm competition and sexual selection (*Kleene, 2005*; *Swanson and Vacquier, 2002*; *Panhuis et al., 2006*).

Against all expectations, however, we find that presence of *Arp53D* may impair rather than enhance male fertility, both in isolation as well as in competition with WT males. This finding is at odds with Arp53D's predominant expression in male testes across *Drosophila* evolution (*Figure 1— figure supplement 1F*) and its localization to specialized actin structures in spermatogenesis (*Figures 2* and *3*). Yet, upon loss of Arp53D, we observe no obvious defects in these actin structures and instead observe an acceleration of sperm production (*Figure 5C, D*). It is possible that Arp53D's absence in sperm leads to increased expression and recruitment of other Arps, such as Arp2/3, or actin regulatory proteins in testes to these germline-specific actin structures. However, this does not explain why *Arp53D* is expressed in testes at all, given that it may be costly to male fertility. One

**Table 5.** *D. melanogaster* transgenics constructed.

| Genetic modification | Chromosomal location | Integrated plasmid backbone | Fly strain injected |
|---|---|---|---|
| CRISPR/Cas9 Arp53D knockout | Chr 2, 53D8, 2R:12661915.12662963 | pHD-attP-DsRed (RRID:Addgene_51019)* | RRID:BDSC_55821 |
| sfGFP-Arp53D | Chr 3, 89E11, 3R:17052863 | p[acman] (*Venken et al., 2006*) | RRID:BDSC_9744 |
| sfGFP-△N-term -Arp53D | Chr 3, 89E11, 3R:17052863 | attB-DsRed† | RRID:BDSC_9744 |
| Nterm Arp53D-sfGFP | Chr 3, 89E11, 3R:17052863 | attB-DsRed† | RRID:BDSC_9744 |
| sfGFP-Nterm Arp53D-Act5C | Chr 3, 89E11, 3R:17052863 | attB-DsRed† | RRID:BDSC_9744 |
| *WT Arp53D* (tagless) | chr2R:16,774,308–16,774,426 | attB-sfGFP‡ | Arp53D KOs isogenized in the Oregon-R background |

*pDsRed-attP is from Melissa Harrison and Kate O'Connor-Giles and Jill Wildonger (Addgene plasmid # 51019; http://n2t.net/addgene:51019; RRID:Addgene_51019).

†Vector encoding an attB site and 3xP3-*DsRed* flanked by loxP sites.

‡Vector encoding an attB site and 3xP3-sfGFP.

possibility is that *Arp53D* may serve to monitor the quality of sperm produced. Under this model, *Arp53D*-KO males may produce more sperm, but of an inferior quality, leading to progressively less fit progeny. These impairments could be subtle and require multiple generations to reveal themselves, like in our population cage experiment (*Figure 6*). Alternatively, Arp53D may confer a fitness benefit to male fertility in untested conditions like the presence of *Wolbachia*. Since a predominant testis-specific expression pattern is a hallmark of *Arp53D* and other non-canonical Arps in *Drosophila* and mammalian species, we favor the possibility of at least a context-specific beneficial role for *Arp53D* in male fertility.

Despite its weak expression outside the male germline, we show that *Arp53D* plays an important beneficial role in embryonic development (*Figure 9*). We find that the fitness defects arising from lack of maternal contribution of *Arp53D* synergize with absence of zygotic *Arp53D* expression in the zygote, leading to more severe embryonic inviability (*Figure 9A*) and reduced number of adult progeny (*Figure 7A*). Therefore, we conclude that *Arp53D* is a maternal-zygotic lethal effect gene, in which embryonic lethality is exacerbated when both maternal and zygotic genotypes are *Arp53D*-KO (*Figure 8C*). We show that lack of *Arp53D* leads to gross nuclear abnormalities and increased nuclear fallout during early embryonic development (*Figure 9B–E*). This may be because Arp53D directly acts upon nuclei during this process. However, we favor the alternative hypothesis that nuclear fallout is an indirect consequence of loss of *Arp53D* and its regulation of the actin cytoskeleton. Many cytoskeletal proteins are critical in cellularization and nuclear migration during early embryogenesis (*Sullivan et al., 1993*). These developmental events are also highly sensitive to heat stress. Actin networks are dramatically reorganized in the heat stress response of *Drosophila* embryos, leading to decreased embryonic viability (*Figard et al., 2019*). We speculate that Arp53D may regulate embryonic actin networks in the heat stress response, explaining why embryonic defects upon *Arp53D* loss are strongly exacerbated at high temperature.

Our studies reveal that, contrary to assumptions based on patterns of highest expression, non-canonical *Arp53D* plays important roles in many aspects of *D. melanogaster* biology beyond male fertility. Numerous genes exhibit highest expression in the testis and brain, two tissues that are especially transcriptionally promiscuous, even though their most important function may manifest elsewhere. The 'out-of-testis' hypothesis predicts that the male germline provides an initial 'gene nursery' for evolutionary innovation, with diversification subsequently broadening its expression profile (*Assis and Bachtrog, 2013*; *Vinckenbosch et al., 2006*; *Nyberg and Carthew, 2017*). Recent studies of histone variants, which were originally thought to be 'testis-specific' in *Drosophila* and mammals, based on RT-PCR data, demonstrated that their expression and function also extends to females (*Kursel et al., 2021*; *Molaro, 2020*). Similarly, *Umbrea,* which is highly testis-enriched, is required for chromosome segregation more broadly (*Ross et al., 2013*). Our findings suggest caution against the practice of using gene expression patterns as a surrogate for function. We conclude that even non-canonical 'testis-specific' Arps like *Arp53D* may, in fact, play surprising roles outside the male germline.

## Materials and methods

### Phylogenetics and positive selection tests

All sequences (*Table 1*) were obtained from Flybase (*Thurmond et al., 2019*) and/or NCBI and aligned using MAFFT (*Katoh and Standley, 2013*) in Geneious (*Kearse et al., 2012*) (RRID:SCR_010519). Nucleotide sequences were used for the maximum likelihood tree generated using PhyML and 100 bootstraps. For positive selection tests, unpolarized MK tests (*McDonald and Kreitman, 1991*) were conducted online (*Egea et al., 2008*) with 198 *D. melanogaster* strains (DPGP3) (*Lack et al., 2015*), obtained from the genome browser Popfly (*Hervas et al., 2017*), and the *D. simulans* reference allele (*Hu et al., 2013*). We manually curated the gene sequence for *D. simulans Arp2*, which was incorrectly annotated in the automated gene prediction model (*Hu et al., 2013*), likely due to poor alignment with *D. melanogaster Arp2*. For all genes, only Zambian strains were used due to having many sequenced strains. Strains whose sequence contained one or more N bases were initially removed; rare polymorphisms (<5% of total sequences) were ignored (*Fay et al., 2001*). After deducing that a stretch of contiguous N bases represents a polymorphic 15 bp deletion rather than poor sequencing quality, we repeated the MK test using all strain sequences.

To assess site-specific positive selection, we generated codon-based alignments of *Arp53D* coding sequences in 10 species in the *D. melanogaster* subgroup using Geneious (*Kearse et al., 2012*). The alignment and corresponding species tree were used in the CODEML algorithm in the PAML suite (*Yang, 2007*) (RRID:SCR_014932) to compare the M7, M8a, and M8 NSsites models. The program determines whether the evolution of *Arp53D* best fits the M8 model, which allows for positive selection, or the M7 or M8a models, which do not allow for positive selection. The difference between the models' log-likelihoods was assessed for statistical significance using a chi-squared test. We used several starting omegas (0.4, 1.0, and 1.5) and codon frequency models (F3x4 and F61), none of which indicated site-specific positive selection in *Arp53D*.

## Sequencing and RT-PCR

To obtain genomic DNA from flies for subsequent PCRs and Sanger sequencing, one or two flies were ground in 10 mM Tris-HCl pH8, 1 mM EDTA, 25 mM NaCl, and 200 µg/mL Proteinase K. The fly lysate was incubated at 37°C for 30 min, followed by 95°C for 3 min to inactivate Proteinase K. Following centrifugation, the supernatant was used for analysis. PCRs were conducted with Phusion according to the manufacturer's instructions (NEB).

To assess Arp53D RNA expression, whole flies (10 minimum) were ground in TRIzol (Invitrogen). Following centrifugation, the supernatant was chloroform-extracted and the resulting soluble phase was isopropanol-extracted to precipitate RNA. RNA was then centrifuged, washed with 75% ethanol, dried, and resuspended in RNAse-free water. Samples were treated with DNaseI (Zymo Research) or TURBO DNase (Thermo Fisher) according to the manufacturer's instructions. DNase-treated samples were then further purified and concentrated using an RNA-cleanup kit (Zymo Research), and cDNA was obtained using SuperScript III first-strand synthesis (Invitrogen). All primers used are listed in *Table 3*. To detect low amounts of Arp53D cDNA from female tissue, a touchdown PCR protocol (*Korbie and Mattick, 2008*) was conducted. The starting annealing temperature was 70°C, which was decreased 0.5°C every cycle for 13 cycles, followed by 17 cycles at 64°C.

## Immunoblot analysis

Approximately 30 testes from the transgenic line *w⁻; sfGFP-△N-term-Arp53D* and the line *w⁻; Arp53D KO; sfGFP-Arp53D* (full length) were dissected separately in PBS and centrifuged. After the supernatant was removed, the pellets of testes were flash frozen. Once thawed for immunoblot analysis, 20 µL of 4X NuPAGE LDS sample buffer (Thermo Fisher) was added to each pellet, which was resuspended and boiled for 5 min at 100°C. Protein samples were loaded on a mini-protean TGX stain-free protein gel (BioRad), run with Tris/Glycine/SDS buffer and transferred to a PVDF trans-blot turbo membrane (BioRad). After blocking with 5% milk in Tris-buffered saline (TBS) and 0.1% Tween-20 (TBST), the membrane was probed with anti-GFP and anti-tubulin in TBST for 1 hr at room temperature, followed by three 10 min washes with TBS. The membrane was then incubated for 45 min at room temperature with IR dye 680 anti-chicken (LI-COR) and/or IR dye 800 anti-rabbit 800 nm (LI-COR) in TBST (see *Table 4* for dilutions). After three final washes with TBS, the membrane was scanned with 680 nm and 800 nm.

## Generation of the *Arp53D*-KO fly line and reinsertion of WT *Arp53D* for rescue

CRISPR/Cas9 was used to knockout *Arp53D* and replace it with *DsRed* to track the *Arp53D*-KO allele. Both guide RNAs were cloned into pCFD4 (RRID:Addgene_49411) (*Port et al., 2014*). Homology arms (1 kb in length) flanking *DsRed* were cloned into pHD-attP-DsRed (RRID:Addgene_51019) (*Gratz et al., 2014*). Guide RNAs were chosen based on optimal efficiency score and no predicted off-targets (http://www.flyrnai.org/crispr2/). The guide RNAs (TCCTGGAAACATGAGCAGCG and TTGGACGGGTGGTTCCGTCT) targeted internally to *Arp53D*, leading to an early stop-codon and removal of the actin fold domain. The CRISPR/Cas9 targets were chosen because they were least invasive to the nearby essential gene *SOD2* and predicted to not alter *SOD2's* transcriptional regulatory elements. The two plasmids for CRISPR/Cas9 were midi-prepped (Takara Bio) and co-injected by BestGene, Inc in stock 55821 from the Bloomington Drosophila Stock Center (RRID:BDSC_55821). BestGene, Inc isolated transformants, crossed out the gene encoding for Cas9, and balanced the modified second chromosome with *CyO*. The *Arp53D*-KO fly line was backcrossed to the same

Oregon-R fly line used in fertility assays for eight generations, sequence-verified, and confirmed for lack of *Arp53D* expression and absence of *Wolbachia* (*Figure 5—figure supplement 1B–F*). The *Arp53D*-KO line was also separately backcrossed to the *w1118* fly line for six generations and sequence verified; this white-eyed line was subsequently used for cytological analyses and population cage experiments. For isogenization, females heterozygous for the *Arp53D*-KO allele were collected in each generation for a subsequent backcross since meiotic recombination only occurs in females, allowing for further mixing of genetic backgrounds. Heterozygous virgin flies were then crossed to obtain a homozygous *Arp53D*-KO fly strain, which was consistently maintained at room temperature and used for fertility assays.

To test for rescue of the *Arp53D*-KO phenotypes, we used site-directed transgenesis and inserted WT *Arp53D* (PCR-amplified from the Oregon-R *D. melanogaster* strain) into the attP site of the *Arp53D*-KO flies that were isogenized in the Oregon-R background. The construct used for transgenesis included the longer WT *Arp53D* allele (containing the polymorphic 5-codon segment) and its upstream intergenic region (~400 bp), which includes *Arp53D*'s endogenous promoter. The transgene was cloned into an attB vector that encoded sfGFP under the control of an eye-specific promoter, which allowed us to track the presence of the transgene (*Table 5*). The construct was midi-prepped and injected by Rainbow Transgenic Flies, Inc Transformants were selected by identifying GFP fluorescence in the eye and were crossed to maintain as homozygous stocks.

## Fly culturing and generation of fly transgenics

All flies were cultured at 25°C on yeast-cornmeal-molasses-malt extract medium. *D. melanogaster Arp53D* was N-terminally tagged with *sfGFP* followed with a 6-aa intervening linker (GGSGGS). This transgene as well as all Arp53D variants (△Nterm-Arp53D, Nterm Arp53D-Actin, and Nterm Arp53D-sfGFP) included *Arp53D*'s upstream intergenic region (~400 bp) for expression under *Arp53D*'s endogenous promoter. All transgenes were cloned into vectors encoding an attB site and *DsRed* under the control of an eye-specific promoter (3XP3). Constructs were midi-prepped (Takara Bio) and injected by BestGene, Inc into BDSC 9744 (*Table 5*). To construct the △N-term Arp53D fly transgenic, amino acids 1–35 of Arp53D were removed. For the *Nterm Arp53D-Actin* fly line, amino acids 1–35 of Arp53D followed by the GGSGGS linker were added N-terminally to *D. melanogaster Act5C*. For *Nterm Arp53D-sfGFP*, Arp53D's N-terminus (aa 1–35) followed by a 6-aa linker (GGSGGS) was added N-terminally to *sfGFP*, replacing Arp53D's actin fold domain (aa 36–411); this construct did not have an N-terminal sfGFP tag. PCR products encoding the tags, linkers, and Arp53D or actin domains were PCR-stitched together and inserted into the vector backbone (*Table 5*) with Gibson technology (*Gibson et al., 2009*) (NEB). Transformants were selected by the eye marker, crossed to *w1118*, and were stably maintained as homozygous stocks. Modified sites were verified by PCR and subsequent Sanger sequencing.

## Immunofluorescence and live imaging

For live and fixed imaging, testes at room temperature were dissected from 0- to 2-day-old males in PBS using a dissecting scope. Live imaging was also conducted to confirm lack of fixation artifacts in immunofluorescence. For live imaging of individual cysts at all stages of spermatogenesis, dissected testes were transferred to a drop of PBS containing Hoechst 33342 (Invitrogen) and SiR-actin, a fluorescent molecule that binds filamentous actin (*Lukinavičius et al., 2014*) (10 µM; Cytoskeleton, Inc) on a slide and pulled apart, evenly distributing visibly elongated cysts. Cysts were stained for 5 min at room temperature, and then a coverslip was placed on top for imaging.

For fixation of individual cysts, cysts were separated (as done for live imaging) in PBS. After a coverslip was placed on the slide, it was submerged in liquid nitrogen. Tissue was fixed with either paraformaldehyde (PFA) or methanol. For PFA fixation, the coverslip was removed from flash-frozen slides and the slides were placed in 100% ethanol for 10 min. Then fixation with 4% PFA in PBS took place for 7 min at room temperature. Tissue was then permeabilized twice for 15 min each with PBS and 0.3% Triton X-100% and 0.3% sodium deoxycholate. Alternative fixation with methanol took place at −20°C for 5 min, followed by incubation in acetone −20°C for 5 min. After both fixation protocols, slides were washed once with PBST for 10 min and then blocked with 3% BSA in PBST for 30 min. Primary antibody incubations took place overnight at 4°C, followed by three 15 min washes in PBS at room temperature. Slides were incubated with secondary antibody for 1 hr, followed by four

15 min washes with PBS. Slides were then either washed once with Hoechst 33342 (Invitrogen) or DNA-stained with mounting media containing DAPI (Thermo Fisher). Following the addition of mounting media, a coverslip was placed and sealed with nail polish. *Table 4* includes antibody dilutions that were used.

To conduct immunofluorescence with whole fixed testes, dissected testes were immediately fixed with 2% PFA in periodate-lysine-paraformaldehyde (PLP) buffer for 1 hr at room temperature, and then permeabilized with PBS with 0.5% Triton X-100 for 30 min. Testes were blocked for 30 min with 3% BSA in PBS plus 0.1% Triton X-100 (PBST). Incubation with primary antibodies took place overnight at 4°C. Testes were then washed several times, followed by secondary antibody incubation for 2 hr at room temperature. After washing three times with PBST, testes were mounted onto slides with VECTASHIELD antifade mounting media with DAPI (Thermo Fisher).

For imaging testes and seminal vesicles under heat stress, virgin males were aged at 29°C for 3 days. Whole testes and the seminal vesicle were then dissected and fixed in 4% PFA in PBS for 25 min. After washing with PBST three times for 15 min, tissue was incubated with 2 µM SiR-actin (*Lukinavičius et al., 2014*) for 3 hr at room temperature (for detection of actin cones). Tissue was then washed with PBS three times for 10 min each, followed by a 5–10 min incubation with Hoechst 33342 (Invitrogen). Testes and seminal vesicles were mounted on slides with VECTASHIELD antifade mounting media.

Embryos were imaged as done previously (*Mavrakis, 2016*). Virgin females and males were collected, and after 5 days of aging at room temperature, crosses were setup at 29°C in cages. Embryos were collected and dechorinated with 50% bleach for 30 s. Embryos were then washed several times with embryo wash buffer (0.7% NaCl, 0.05% Triton X-100) and then fixed in 4% PFA with heptane (1:1 ratio) for 25 min. The PFA (bottom layer) was removed and an equal volume of methanol was added to remove the vitelline envelope. The embryos were vortexed vigorously for 30 s, and after removing the supernatant, they were washed several times with 100% methanol and stored at −20°C. To probe for DNA, embryos were first rehydrated in PBST and blocked with 10% BSA in PBS for 10 min. Following a 10 min incubation with Hoechst 33342 (Invitrogen), embryos were mounted on slides with VECTASHIELD antifade mounting media. All live and fixed samples were imaged using a confocal microscope (Leica TCS SP5 II) and LASAF software (Leica).

## Fertility assays

Oregon-R flies were used as WT flies because *Arp53D* KOs were isogenized in the Oregon-R genetic background. Female and male virgins were collected for all assays. Females were 1–5 days old, and males were 1–2 days old. Crosses were setup with females in excess (female:male ratios of 5:2 or 10:3, unless noted otherwise), and matings took place for a week at 25°C or 29°C with vials flipped every 2–3 days. Light/dark cycles were maintained consistently. For heat-stress experiments, virgins were maintained at 25°C until crosses were setup and then transferred to 29°C. All adult progeny were quantified on the last possible day before emergence of progeny from the next generation. With day 1 being the time at which crosses were setup, day 15 or 16 was the last day the first generation could be counted at 25°C; day 12 or 13 was the last day for 29°C experiments. For quantification of fluorescence (the *Arp53D*-KO allele) in the Oregon-R background, DsRed fluorescence was visualized in the ocelli because pigmentation obscured fluorescence in the eye. Heterozygous flies were generated by crossing KO females to Oregon-R flies at room temperature. Homozygous *DsRed* flies were denoted by strikingly fluorescent ocelli and dim fluorescence of the body, whereas the ocelli of heterozygous flies were dim and required close observation to differentiate from WT flies. All fertility assays were conducted at least twice. Parental flies that died in all crosses were tallied and did not differ significantly among genotypes.

To examine the development of embryos, female and male virgins were collected as for the fertility assays. Fly crosses were setup with females in fourfold excess and allowed to lay at 29°C for 2 hr. The eggs laid were counted and then returned to 29°C. After 24 hr, unhatched eggs were quantified. To compare the proportion of laid eggs that are fertilized, eggs were collected after 2 hr of laying at 29°C and washed in embryo wash buffer (EB, 0.7% NaCl, 0.05% Triton X-100), followed by dechorination with 50% bleach for 30 s. The dechorinated eggs were then washed several times with EB and once with water. They were mounted on an adhesive solution resulting from double-sided tape soaked in n-heptane. Eggs were then covered with Halocarbon oil 700 (Sigma-Aldrich) to prevent

dehydration and imaged with brightfield microscopy (Leica microscope model DMIL LED) to assess cellularization, the first stage of embryogenesis and sign of successful fertilization.

For knockdown of *Arp53D*, RNAi line 108369 (VDRC, RRID:SCR_013805) was used and sequence-verified (as done in *Green et al., 2014*) for integration at the chromosomal 30B site and not the 40D site, which has a non-specific phenotype (*Green et al., 2014*). The line was crossed to *topi-Gal4* flies (generously given by the labs of Lynn Cooley and Christian Lehner) for knockdown in late spermatogenesis.

### Population cage experiment and fitness modeling

The isogenized *Arp53D*-KO line in the *w1118* background was used due to ease of DsRed detection in the eye (as opposed to the ocelli in the Oregon-R *Arp53D*-KO background). Virgin females and males were collected from the *w1118* fly line and the *Arp53D*-KO fly line isogenized in the *w1118* background. Crosses with 50 *Arp53D*-KO females, 25 *Arp53D*-KO males, and 25 *w1118* males were setup in bottles with three replicates. Crosses were passaged every 2 weeks at room temperature. At each passage, 50 females and 50 males were randomly collected without fluorescence detection and without selection based on virgin status. These 100 flies were placed in a fresh bottle, and the remaining progeny were frozen for subsequent detection of DsRed fluorescence. After 1 week of laying before the next generation hatched, the 100 flies were removed and frozen to include in the previous generation's quantification.

In order to gain insight into the fitness differences between *Arp53D*-KO and WT flies, we modeled the population cage experiment (*Figure 6—figure supplement 1*) *in silico*, simulating experimental evolution using a large number of different fitness parameters (https://github.com/jayoung/Arp53D_popCage; *Young, 2021*; copy archived at swh:1:rev:52ff682daab06ba677f43a49de6f5b-d8a0c54a62). Our modeling assumes a freely mating population of infinite size. We defined fitness coefficients for each genotype ($F_{WT}$, $F_{het}$, $F_{KO}$) relative to WT homozygous flies ($F_{WT} = 1$). We explored fitness coefficients for KO homozygous flies ($F_{KO}$) that ranged between 0.4 and 1 in increments of 0.001. We explored three possibilities for heterozygote fitness, where fitness matched either WT ($F_{het} = F_{WT}$), or KO homozygotes ($F_{het} = F_{KO}$), or was exactly intermediate in fitness between WT and KO homozygotes ($F_{het} = (F_{WT} + F_{KO})/2$). We seeded all models using the same genotype combinations as the actual experiment (100% KO homozygous females, and a 50:50 mix of WT homozygous and KO homozygous males). At each generation, we calculated the fraction of randomly selected mating pairs that represented each possible genotype combination ($P_{mat \times pat}$). For each combination of mating pair genotypes, we used Mendelian segregation to determine the fraction of offspring genotypes ($O_{WT}$, $O_{het}$, $O_{KO}$). To obtain the overall fraction of progeny genotypes from all parental genotype combinations, we summed the product of those frequencies ($P \times O$) for all mating pair combinations. After obtaining initial progeny genotype frequencies in each generation, we applied fitness coefficients, multiplying the genotype frequencies by $F_{WT}$, $F_{het}$, $F_{KO}$, and renormalizing genotype frequencies to sum to 1. This strategy oversimplifies the true biology as it applies fitness coefficients only to individual genotypes at each generation, regardless of parental genotypes that we know have strong effects. We iterated these steps over 35 generations and recorded genotype frequencies at each generation. In order to determine which model best fit the data, we calculated the mean absolute error (MAE) for each model (by subtracting the modeled value at the corresponding generation from each real datapoint, taking the absolute value, and then calculating the mean) and selected the model that minimized MAE.

## Acknowledgements

We thank Grace Yuh Chwen Lee for help with the initial McDonald-Kreitman analyses and Ching-Ho Chang for help with RNA-seq analysis. We thank Susan Parkhurst and Barbara Wakimoto for providing fly lines and technical advice and Akhila Rajan for allowing us to use her 29℃ room for critical experiments. We also thank Mollie Manier for offering dissection protocols and helpful discussions about *Arp53D*-KO phenotypes, Leslie Vosshall for providing anti-Androcam antibodies, Kathleen Beckingham for discussing potential co-localization of Androcam and Arp53D, and Lynn Cooley and Christian Lehner for providing the topi-Gal4 fly line. Lastly, we thank Ching-Ho Chang, Lisa Kursel, Antoine Molaro, and Pravrutha Raman for providing comments on the manuscript as well as the rest of the Malik lab for useful discussions and especially Aida de la Cruz for *Drosophila* training. This

work was funded by the Jane Coffin Childs Memorial Fund (CMS), an NIGMS K99 Pathway to Independence Award 1K99GM137038-01 (CMS), NIGMS grant R01GM074108 (HSM), and the Howard Hughes Medical Institute (HSM). HSM is an Investigator of the Howard Hughes Medical Institute.

## Additional information

### Funding

| Funder | Grant reference number | Author |
|--------|------------------------|--------|
| Jane Coffin Childs Memorial Fund for Medical Research | Postdoctoral fellowship | Courtney M Schroeder |
| National Institute of General Medical Sciences | K99GM137038 | Courtney M Schroeder |
| National Institute of General Medical Sciences | R01GM074108 | Harmit S Malik |
| Howard Hughes Medical Institute | | Harmit S Malik |

The funders had no role in study design, data collection and interpretation, or the decision to submit the work for publication.

### Author contributions

Courtney M Schroeder, Conceptualization, Data curation, Formal analysis, Supervision, Funding acquisition, Validation, Investigation, Visualization, Methodology, Writing - original draft, Project administration, Writing - review and editing; Sarah A Tomlin, Data curation, Formal analysis, Supervision, Investigation, Methodology, Writing - review and editing; Isabel Mejia Natividad, John R Valenzuela, Data curation, Formal analysis, Investigation; Janet M Young, Software, Formal analysis, Validation, Investigation, Visualization, Methodology, Writing - original draft, Writing - review and editing; Harmit S Malik, Conceptualization, Supervision, Funding acquisition, Visualization, Methodology, Writing - original draft, Project administration, Writing - review and editing

### Author ORCIDs

Courtney M Schroeder ⬚ https://orcid.org/0000-0002-4526-8321
Janet M Young ⬚ http://orcid.org/0000-0001-8220-8427
Harmit S Malik ⬚ https://orcid.org/0000-0001-6005-0016

### Decision letter and Author response

Decision letter https://doi.org/10.7554/eLife.71279.sa1
Author response https://doi.org/10.7554/eLife.71279.sa2

## Additional files

### Supplementary files

• Source data 1. Raw images for figure supplements.

• Supplementary file 1. The data for *Figure 5B* and *Figure 8A, B* are displayed in individual sheets in the Excel file. For each panel, it is shown how the chi-squared test was conducted and the percent genotypes are graphed per replicate.

• Transparent reporting form

### Data availability

All data are displayed in the main and supplementary figures. Source data files are provided for Figure 5B and Figure 8A-B. The code developed to model the population cage experiment is available on GitHub (https://github.com/jayoung/Arp53D_popCage/releases/tag/v1.0.0).

The following previously published datasets were used:

| Author(s) | Year | Dataset title | Dataset URL | Database and Identifier |
|---|---|---|---|---|
| Luo S, Zhang H, Duan Y, Yao X, Clark AG, Lu J | 2020 | Co-evolution of transposable elements and piRNAs in *Drosophila melanogaster* | https://www.ncbi.nlm.nih.gov//bioproject/PRJNA309630 | NCBI BioProject, PRJNA309630 |
| Rogers RL, Shao L, Sanjak JS, Andolfatto P, Thornton KR | 2014 | *D. melanogaster* reference RNA-seq virgin male carcass | https://www.ncbi.nlm.nih.gov/bioproject/PRJNA257287 | NCBI BioProject, PRJNA257287 |
| Rogers RL, Shao L, Sanjak JS, Andolfatto P, Thornton KR | 2014 | D. yakuba reference RNA-seq virgin male carcass, paired end reads | https://www.ncbi.nlm.nih.gov/bioproject/?term=PRJNA196536 | NCBI BioProject, PRJNA196536 |
| Rogers RL, Shao L, Sanjak JS, Andolfatto P, Thornton KR | 2014 | *Drosophila ananassae* Transcriptome or Gene expression | https://www.ncbi.nlm.nih.gov/bioproject/?term=PRJNA257286 | NCBI BioProject, PRJNA257286 |
| Ma S, Avanesov AS, Porter E, Lee BC, Mariotti M, Zemskaya N, Guigo R, Moskalev AA, Gladyshev VN | 2018 | Comparative Transcriptomics across 14 *Drosophila* Species Reveals Signatures of Longevity | https://www.ncbi.nlm.nih.gov/bioproject/?term=PRJNA414017 | NCBI BioProject, PRJNA414017 |
| Nozawa M, Onizuka K, Fujimi M, Ikeo K, Gojobori T, National Institute of Genetics | 2016 | mRNA-seq of *Drosophila* miranda, D. pseudoobscura, and *D. obscura* | https://www.ncbi.nlm.nih.gov/bioproject/?term=PRJDB4576 | NCBI BioProject, PRJDB4576 |
| Yang H, Jaime M, Mahadevaraju S, Polihronakis M, Oliver B | 2017 | RNA-seq of sexed adult tissues/body parts from eight *Drosophila* species | https://www.ncbi.nlm.nih.gov//geo/query/acc.cgi?acc=GSE99574 | NCBI Gene Expression Omnibus, GSE99574 |
| Yang H, Jaime M, Kanegawa K, Kaneshiro K, Oliver B | 2016 | Expression profiling Hawaiian *Drosophila* species, tissues, and sexes | https://www.ncbi.nlm.nih.gov/geo/query/acc.cgi?acc=GSE80124 | NCBI Gene Expression Omnibus, GSE80124 |
| University of California, Davis | 2020 | Adult *Drosophila melanogaster* testis RNA sequencing | https://www.ncbi.nlm.nih.gov/bioproject/PRJNA613134 | NCBI BioProject, PRJNA613134 |
| University of Rochester | 2019 | Transcriptome sequencing of *Drosophila* simulans clade | https://www.ncbi.nlm.nih.gov/bioproject/?term=PRJNA541548 | NCBI BioProject, PRJNA541548 |
| Oliver B | 2011 | RNA-Seq of Gonads and Carcasses in D. simulans and D. pseudoobscura | https://www.ncbi.nlm.nih.gov/geo/query/acc.cgi?acc=GSM775504 | NCBI Gene Expression Omnibus, GSE31302 |
| Malone JH, Artieri CG, Sturgill D, Zhang Y, Oliver B | 2011 | mRNA-Seq of whole flies from *Drosophila* | https://www.ncbi.nlm.nih.gov/geo/query/acc.cgi?acc=GSE28078 | NCBI Gene Expression Omnibus, GSE28078 |
| University Of California, Davis | 2013 | The population genetics of De novo genes in *Drosophila* | https://www.ncbi.nlm.nih.gov/bioproject/PRJNA210329 | NCBI BioProject, PRJNA210329 |
| Cornell University | 2017 | D. americana, D. lummei, D. novamexicana, and D. virilis: genomes and male transcriptome sequencing and assembly | https://www.ncbi.nlm.nih.gov/bioproject/PRJNA376405 | NCBI BioProject, PRJNA376405 |
| Cornell University | 2019 | Platinum *Drosophila* | https://www.ncbi.nlm. | NCBI BioProject, |

| | | | | | |
|---|---|---|---|---|---|
| | | | genomes | nih.gov/bioproject/PRJNA554780 | PRJNA554780 |
| Celniker SE, Dillon LAL, Gerstein MB, Gunsalus KC, Henikoff S, Karpen GH, Kellis M, Lai EC, Lieb JD, MacAlpine DM, Micklem G, Piano F, Snyder M, Stein L, White KP, Waterston RH, modENCODE Consortium | | 2009 | Transcriptional profile of *D. melanogaster* tissues, stranded RNA-Seq, modENCODE | https://flybase.org/reports/FBlc0000206.html | Flybase, FBlc0000206 |
| Hervas S, Sanz E, Casillas S, Pool JE, Barbadilla A | | 2017 | PopFly: the *Drosophila* population genomics browser | http://popfly.uab.cat | PopFly, PopFly |
| Jambor H, Surendranath V, Kalinka AT, Mejstrik P, Saalfeld S, Tomancak P | | 2015 | The Dresden Ovary Table | http://tomancak-srv1.mpi-cbg.de/DOT/main | The Dresden Ovary Table, DOT/main |

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
