## [Decision Letter]

[Editors' note: this paper was reviewed by Review Commons.]

**Acceptance summary:**

This manuscript was reviewed through the Review commons process by three well -respected reviewers in the field. The reviews were largely positive with several detailed suggestions made for improvement. The authors took these suggestions to heart and provided a well-documented and honest summary of each result. What makes this paper particularly suited for a journal like *eLife* is its universal acclaim of novelty and high significance. The result that a protein highly expressed in the testis has been selected over evolution, not so much because it contributes to sperm function, but because of expression elsewhere, is, to say the least, extremely surprising but has impressed the reviewers. To me, the fact that this protein is an Arp, with binding function to actin makes this finding even more unusual and interesting. The work will generate discussions amongst evolutionary biologists for sure.

---

## [Author Response]

We would like to thank the three reviewers for their time and thoughtful comments on the manuscript. We found their constructive feedback extremely helpful and are grateful for the peer review process in clarifying and strengthening our study. In response to their comments and suggestions, we have extensively revised the Discussion section of our manuscript. We have also added four new experiments in our revision. We summarize them here briefly and describe them in more detail in our revised manuscript and in the point-by-point rebuttal.

– Introduction of *Arp53D* rescue transgene in place of the *Arp53D*-KO allele: Although this rescue transgene is not expressed as strongly as the endogenous gene, we find it nonetheless robustly rescues the embryonic viability phenotype, but does not significantly lower male fertility.

– Extensive population-genetic modeling of our population cage experiments: These analyses revealed that we had grossly underestimated the selective coefficients associated with wildtype *Arp53D* alleles in our previous submission. Instead of a 2-3% effect, we now demonstrate that wildtype alleles have a 30-40% selective advantage over the KO allele.

– Cytological analysis of the basis of male fertility increases in *Arp53D*-KO flies: We show that although a static readout of total sperm (via seminal vesicle size) is unchanged, there is an increased number of individualizing cysts in *Arp53D*-KO flies, suggesting that sperm production is accelerated.

– Cytological analysis of the basis of embryonic inviability in *Arp53D*-KO embryos: We show that there is an increased number of disorganized nuclei and a higher propensity of ‘nuclear fallout’ in *Arp53D*-KO embryos at 29ºC, suggesting that Arp53D plays a key role in helping facilitate early divisions in *Drosophila* embryos under heat stress.

Some of these additional experiments were carried out by a new colleague, Isabel Mejia Natividad, who is now added as an author in the revision.

Reviewer #1 (Evidence, reproducibility and clarity (Required)):Reviewer summary:In this manuscript, the authors characterize the orphan actin related protein Arp53D, which is conserved in all *Drosophila* species with available full genome sequences but not in other dipterans. Sequence analysis of Arp53D in ~200 D. melanogaster populations and in *D. simulans* suggests that this gene is experiencing positive selection in melanogaster, implying that it contributes positively to fitness in some way. The authors corroborate this observation using a population study, which reveals that the allele frequency of an Arp53D knockout is strongly and reproducibly reduced within 15-20 generations. Because publicly available RNA-seq data show this gene to have highest expression in the *D. melanogaster* testis, the authors begin by observing protein localization there. Consistent with its close homology to actin, Arp53D localizes to two actin-enriched structures in male germ cells, fusomes and actin cones of the individualization complex, as revealed by a GFP-tagged transgene. Deleting Arp53D's unique N-terminal region prevents this localization, and attaching the N-terminal region to canonical actin drives concentrated localization to these structures, indicating that the N-terminal region is necessary and sufficient for Arp53D's specific localization in the male germline. However, knocking out the gene does not perturb gross fusome or individualization complex structure, and, surprisingly, leads to increased male fertility. Consistently, knockout males apparently have an advantage in a sperm competition assay. Even more surprisingly, given the gene's low expression in females and early embryos, knockout leads to reduced embryonic survival, with both maternal and zygotic contributions to the effect. Altogether, the data show that despite Arp53D's strong expression and specific localization in the testis, it actually is detrimental there; nevertheless, it confers a survival advantage in embryos, with both maternal and zygotic effects, despite minimal maternal expression.

We thank the reviewer for a clear summary of our findings.

Major comments:1. Locus specificity of the male fertility and embryonic viability phenotypes must be substantiated before the manuscript is ready for publication. The authors should be able to use their stock carrying the Arp53D knockout with the tagged Arp53D transgene to demonstrate rescue of the major knockout phenotypes.

We thank the reviewer for raising this important point, which was also raised by other reviewers. We have now added experiments to confirm the locus specificity of the phenotypes in the Oregon-R *Arp53D*-KO strain background used for all genetic experiments. To create the *Arp53D-*KO strain, we had previously knocked in *DsRed* with an attp site, which allows for site-directed transgenesis, followed by repeated backcrossing with Oregon-R flies. In our revision, we took advantage of this attP site to integrate an untagged wild-type (WT) *Arp53D* gene and its upstream endogenous promoter into this genetic location. Our approach is now outlined in the new Figure 5E. This rescue allele robustly rescued the reduction in progeny count seen with the female knockout (new Figure 7C) confirming that the embryonic viability phenotype is a direct result of *Arp53D*-KO and not a linked mutation. However, male fertility was only slightly decreased in this transgene rescue without statistical significance (Figure 5F). We attribute this slight decrease in male fertility to dosage of *Arp53D* expression. Indeed, we find that the ‘rescue’ *Arp53D* transgene is expressed less than WT *Arp53D* in Oregon-R (Supplemental Figure S5I), which appears sufficient to rescue embryonic viability but not lower male fertility. These rescue experiments confirm our surprising findings that *Arp53D*’s evolutionary retention is predominantly due to its role in embryonic development.

2. The manuscript would be enhanced greatly by highlighting in the discussion how the key finding, that despite Arp53D's strong expression in the testis its positive selection appears to be driven by a somatic function in the embryo, deviates from expectations under the hypothesis that the testis is often a permissive site for gene evolution (see e.g., Kondo et al. 2017, Assis and Bachtrog 2013, Zhao et al. 2014, Kaessmann 2010). It also is worth mentioning that this story provides a cautionary lesson against the practice of using gene expression patterns as proxies for function. Replacing the current conclusion sentences with some discussion along these lines would emphasize the strong impact of the results (see point 13 below).

We thank the reviewer for giving us the opportunity to clarify these important points. We were indeed surprised by the pertinent phenotype being unrelated to the predominant site of expression of this gene. We have added to the discussion to caution against the practice of using gene expression patterns as proxies for function.

Minor comments:1. The title is vague and does not highlight the key finding of the paper well. I suggest emphasizing Arp53D's unexpected negative effect on male fertility or positive effect in the embryo, despite its striking male specific expression.

We agree with this suggestion and have changed the title to “An actin-related protein that is most highly expressed in *Drosophila* testes is critical for embryonic development.”

2. The Abstract should be edited at lines 35-37, with bracketed words removed: *"Drosophila* and mammals also encode divergent non-canonical Arps [in their male germline] whose roles remain unknown. Here, we show that Arp53D, a rapidly-evolving *Drosophila* Arp, localizes to fusomes and actin cones, two [male] germline-specific actin structures critical for sperm maturation, via its non-canonical N-terminal tail." The gene is encoded in all cells, and fusomes are found also in the female germline.

We thank the reviewer for these suggestions, which we have gratefully accepted.

3. Results line 111: How were Arp53D orthologs distinguished from other Arp paralogs? What criteria were used? Please add more detail to the Methods section, as Arps are notoriously difficult to distinguish by sequence alone (see Muller et al. 2005).

Arp53D orthologs were distinguished from other Arp paralogs by phylogeny and shared synteny. We have now clarified this in our revision.

4. Results line 135, remove bracketed word: "The MK test compares the ratio of [fixed] non-synonymous (amino acid replacing, PN) to synonymous (Ps) polymorphisms within a species…"

We thank the reviewer for pointing out this error, which we have corrected.

5. Results lines 156-159: The conclusion that inter-species substitutions are concentrated at the N-terminus of Arp53D is not justified. The cartoon in Figure S1E shows only four out of more than 20 in the unstructured N-terminal region.

We agree with this comment and have edited this section accordingly.

6. Results lines 160-162: Please provide more explanation of this analysis and its meaning.

We have added an explanation as follows:

“These McDonald-Kreitman results revealed that *Arp53D* evolved under positive selection in recent evolutionary time (since *D. melanogaster* and *D. simulans* divergence) but do not pinpoint which residue changes were functionally important. […] This suggests that the signature of positive selection does not recur in the same subset of residues.”

7. Results lines 173-175: Please clarify in the text and/or legend where the RNA-seq data came from. Are they publicly available ModENCODE data pulled from Flybase?

Thank you for pointing out this inadvertent omission. The tissue-specific RNA-seq data for *D. melanogaster* was ModENCODE data obtained from Flybase, which is now explicitly acknowledged in the revision along with citations to the RNA-seq data from non-*melanogaster* species.

8. Results lines 209-212, revise text: "During late stages of spermatogenesis, spermatids must separate and obtain their own individual membranes. This process is carried out by the individualization complex containing 64 'actin cones', each of which is a cone of actin filaments, which forms around each sperm head when nuclear condensation is complete. Cones translocate along the axonemes of the sperm tails to push out excess cytoplasm ("cystic bulge") while encasing each sperm in its own membrane."

Thank you for pointing out the lack of clarity. We have rewritten this to ensure clarity.

9. Results line 220: Remove the word "punctate," as this word indicates the presence of puncta, which are not seen in Figure 3C and 3D. SfGFP-Arp53D puncta do seem to appear in Figure 3B, along the axonemes caudal to the actin cones, but this is not described in the text.

We have clarified this point in our revision. We agree with the reviewer’s suggestion to remove the word “punctate”. To clarify, we did observe small puncta along the axonemes; however, this is not always seen with live imaging. We now describe these observations more clearly in our revision.

10. Results line 224: Describing the leading edge of the actin cones as "the site of active actin polymerization" is misleading, because actin polymerization happens in the rear bundles as well (see Noguchi et al. 2008). Also, the phrase "membrane extrusion" should be changed to "cytoplasm extrusion."

We thank the reviewer for raising this point. We have edited this line in the revision to describe the actin cone’s leading edge as “the site of active cytoplasm extrusion.”

11. Results line 227: According to the images presented in Figure S2D, Arp53D localizes adjacent to Androcam rather than colocalizing.

We thank the reviewer for this close observation, and we agree that the image suggests they are adjacent. However, previous literature shows that Androcam extends to the very tip of actin cones similar to what we see with Arp53D (citation included in revision). In our images, Androcam’s apparent lack of direct overlap with Arp53D may be due to steric competition between the antibodies probing for Androcam and Arp53D, or the much weaker staining of Androcam, making Arp53D appear more prominent at the leading edge. We have replaced the insets with a focus on cones exhibiting slightly better staining of Androcam, and we have re-worded the text to state the proteins are proximal, rather than definitively co-localizing. We have also noted in the revision that Androcam staining extends beyond where Arp53D is concentrated.

12. Results line 234, remove bracketed words: "It first localizes to the fusome [as it forms] during meiosis…"

We agree with and have made this edit suggestion.

13. Discussion: The last two lines are abstruse- please replace them (see above).

We agree with and have made this edit and suggestions made by the reviewer in Major comment #2.

14. Methods lines 651-654: Please provide more details on how the cloning was accomplished.

We apologize for the brief description in our earlier submission. In our revision, we have now included a more detailed description of the cloning details and strategy.

15. Figure S3B: The doublet on the blot is described in the figure legend as degradation, but it could be the result of processing or post-translational modifications.

We agree. We now note this suggested possibility in the revision.

16. Figure S3D: Can these data be incorporated into main Figure 4? They emphasize the striking redistribution of canonical actin away from its normal localization by the Arp53D N-terminal region.

We thank the reviewer for this suggestion, which we have gratefully accepted, incorporating these images into a revised Figure 4.

17. Figure S3E: Please include a blot for Nt-GFP to show protein expression at the expected size, as the fluorescent signal is weak. Also, this panel is mislabeled in the figure legend.

We are grateful to the reviewer for noting our mislabeling, which we have now corrected. To verify the expression of Nt-GFP, we conducted an immunoblot analysis with testis lysate, which showed the expression of Nt-GFP at the expected size and at comparable levels to that of GFP-tagged full-length Arp53D expressed in testes (now included in Supplemental Figure S3E).

18. Figure 5D and Figure 6E,H: When experimental data are compared to expected values, use a Chi squared test rather than ANOVA to determine significance.

We appreciate this comment. We have now corrected and used the appropriate test for these panels (now Figure 5B and Figure 8A-B) and related experiments. We also include the raw data used for the chi-squared tests in Supplementary File 1.

19. Figure S5E-F: Please include number of eggs counted for each experiment.

As requested, we now report the number of eggs counted for each panel in the figure legend in our revision (now Supplementary Figure S8B-C).

Reviewer #1 (Significance (Required)):Significance: In addition to highly conserved conventional actins, all eukaryotes possess numerous actin paralogs that together compose a large actin superfamily, which includes many sub-families of actin related proteins (Arps) that also are largely well conserved. Arps carry out a wide array of functions, both in the cytoplasm and the nucleus, investigations of which have focused mostly on well conserved family members and largely have neglected orphan Arps, which are not readily assigned to sub-families. This manuscript investigates a previously uncharacterized orphan Arp that is highly expressed in *Drosophila* testes. The authors find that despite its striking localization to male germline actin structures, it is detrimental to male fertility in *D. melanogaster*. The findings suggest that its apparent positive selection is attributable to embryonic functions, implying a developmental tradeoff between germline and somatic development. The findings are surprising in light of a prominent evolutionary model suggesting that high expression of paralogs and/or new genes in the testis permits evolution of new functions there without compromising preexisting somatic functions of the gene family. This novel investigation into a divergent Arp paralog with strong expression in the testis but an apparent detrimental effect there will likely be of great interest to evolutionary biologists.

We are grateful for this positive and complimentary summary by the reviewer. We also note that our study is the first detailed functional dissection of an ‘orphan’ Arp paralog. Our findings suggest that despite being lineage-specific, such Arps encode very important functions.

Reviewer keywords: *Drosophila,* spermatogenesis, germline actin dynamicsReviewer #2 (Evidence, reproducibility and clarity (Required)):Summary:Schroeder et al. investigate the role of a rapidly-evolving actin paralog in *Drosophila.* This gene, Arp53D, is conserved over 65MY in the Drosophila lineage, but is not found outside it. Arp53D's highest expression is in the testis, and the protein is seen to localize with two actin-rich structures, the fusome and the actin cones, though not always exactly with actin. Despite its conservation and localization, Arp53D appears to have a negative effect on male fertility, as its knockout and knockdown males are less fertile than control. However, its activity is needed for optimal fertility because of a maternal effect on survival or development. That effect can be partially rescued by zygotic expression.

We thank the reviewer for accurately summarizing our findings.

Major comments:Is the study convincing, are the experiments replicated, statistics adequate, is there enough information to reproduce?:Yes. This is an excellent paper in all respects. It reports a new and interesting finding. The experiments are beautifully done and rigorous. The data are convincing. The question is addressed from several perspectives. The paper is well written, well argued, and gives complete details.

We thank the reviewer for the very positive comments and constructive suggestions for improvements, which we have now incorporated into our revision.

I have some requests for information that seems important for interpretation. I believe these are straightforward and realistic to do.1. Does the sfGFP-Arp53D rescue the defects of Arp53D knockout? Otherwise, it is not completely certain that its localization reflects where the endogenous protein is, or is needed.

This is a fair point. We chose to tag Arp53D with sfGFP at the N-terminus because actin proteins can only be tagged at the N-terminus in *Drosophila*; a C-terminal tag disrupts protein polymerization (Brault *et al.*, 1999). However, we believe it is unlikely that this tag alters or dictates localization because actin’s function is not altered with an N-terminal tag. Moreover, we see that actin localizes only to cones and the fusome upon addition of Arp53D’s N-terminus (Figure 4), similar to tagged full-length Arp53D.

We did not attempt to rescue with the sfGFP-Arp53D line because the genetic background is different from the *Arp53D*-KOs that were used in all our genetic experiments. Since fertility levels can vary strain to strain, it is important to have a WT control that is closest in genetic background, which we lack for the sfGFP-Arp53D line. This transgenic fly line has a mix of the *w1118* background and the background of the original injected fly line (see Table 5).

2. Sperm counts would provide an easy way to determine whether the mutant’s problem is in sperm production, mating/transfer, or in function of the sperm themselves.

This is a valuable suggestion. In our revisions, we include additional cytological analyses to assess the basis of male fertility differences. We now report that *Arp53D-KO* testes have more cysts with actin cones than WT in the Oregon-R background, suggesting accelerated sperm production. However, we do not see significant evidence for increased size of the seminal vesicle.

Interpretation suggestion:1. Although the “gene nursery” model might imply that Arp53D was selected on for a testis function, I wonder whether it might actually been selected for its beneficial embryo function, and later acquired testis expression. Its testis function might be a tolerated level of competition between it and a different Arp that localizes to fusomes or actin cones. In this latter scenario, if you get rid of all or some of Arp53D from the testis, that might allow the other Arp to do its job unimpeded, thereby improving male fertility.

We are grateful to the reviewer for suggesting this excellent interpretation. We do believe Arp53D has been evolutionarily retained for the beneficial embryo function in the long term. Since Arp53D does not appear to negatively impact male fertility, we also now suggest the possibility that another Arp, such as Arp2/3 or a different actin regulatory protein, may in fact be playing a role that masks a defect in Arp53D’s absence. Together with a recommendation made to further comment on the gene nursery model by Reviewer 1 (major comment #2), we have revised our Discussion extensively to incorporate these suggestions.

Minor comments, easily addressable:Prior studies are referenced and considered.Overall text, and figures, are clear and accurate, but there are some wording imprecisions that would be good to correct (by line number):2: Arp53D is not “(an) actin”. It is an Arp. The rest of the paper, including the running title, refers to it correctly.

We now refer to Arp53D as an Arp rather than an actin in the revised title.

36-37 and 193-194: The authors correctly note later in the paper that fusomes are found in both male and female germlines. But the wording in these lines (e.g. “fusomes and actin cones, two male germline-specific…”) implies that fusomes are only in male germlines. Please reword.

We agree with this comment and have removed the word “male” describing the germline-specific structures to avoid this confusion.

44-45: seems too strong a conclusion. You don’t know what has led to its longterm retention.

We agree with this comment. In our revision, we suggest that embryonic function most likely explains why Arp53D has been evolutionarily retained. In the discussion, we explain there may be additional functions, including in the male germline, that have contributed to retention.

112: did you look for what was in the Arp53D-lacking genomes in that place, by synteny? Can you get info on how Arp53D suddenly appeared there?

Unfortunately, the age of *Arp53D* and the rapid evolution of *Drosophila* genomes makes detailed inferences difficult, as we now highlight in our revision:

“A previous study proposed that *Arp53D* arose from retroduplication of *Act88F*, which encodes a *Drosophila* muscle actin^24^. […] The ancient evolutionary origin of *Arp53D* does not allow us to determine whether *Arp53D*’s unique N-terminus was acquired from the intergenic DNA sequence upon retroduplication or via subsequent insertions after retroduplication.”

124: do other genes have regions similar to this N-term extension, or is it Arp-specific as the sentence seems to imply?

As we indicated above, we have found no homology between the N-terminal region of Arp53D to any coding or non-coding sequence in any *Drosophila* (or other) genome.

239-240: seems too strong. The data don't show that Arp53D carries out specialized roles at cones and fusomes.

We agree and have edited this statement to reflect the possibility of specialized roles at germline structures.

253: do you know that the sfGFP-deletion fusion is as stable as sfGFP-Arp53D? If not, that could also give rise to what you saw.

In our revision, we considered the possibility that the Arp53D actin-like domain without the N-terminus may be less stable. However, we find that this domain is likely as stable as actin and full-length Arp53D based on Western blot analyses.

259: suggest adding: 'within our detection limits' at the end of the sentence.

We have added the suggested clause.

280: I think it is too strong to conclude that the interaction requires an actin fold. When comparing N-term-SfGFP to sfGFP fused to all of Arp53D I think you can only conclude that the localization requires sequences in Arp53D beyond the N-terminus. Not a particular motif.

We agree with this comment. In our revision, we now propose that Arp53D’s localization requires sequences in the actin domain or the tertiary structure of the actin domain.

294: I would find it clearer as:.… changes in SOD2, an essential gene that is located upstream of Apr53D.

We thank the reviewer and have edited the sentence as suggested.

301: in the reverse of my interpretation-suggestion, any lack of phenotype of the KO could reflect some redundancy among Arps.

We agree this may be the case in males (also suggested by Reviewer #1) and have added this possibility in our revision.

413: I did not understand your phrase "the maternal source of Arp53D is unclear". Doesn't the previous RNAseq data detecting it in ovaries, in combination with the maternal effect, suggest that it is expressed egg chambers?

We apologize for this awkward phrasing. We were trying to make the same point as the reviewer that the maternal effect suggests that Arp53D must be expressed in egg chambers or somatically required for some aspect of egg maturation and post-fertilization development, yet we and others find only weak levels of expression. We now include additional RT-PCR analyses of Arp53D in dissected ovaries to confirm that Arp53D is indeed expressed, albeit weakly, in ovaries (Supplementary Figure S7B).

699: suggest rewording to: For heat-stress experiments…

We have changed the wording as suggested.

Reviewer #2 (Significance (Required)):A new and significant finding from several perspectives: the role and localization of a newly-evolved Arp, the evolutionarily-interesting case of a gene that is beneficial in some contexts and not in others. A strong and new addition to the exciting field of gene birth/death.

We thank the reviewer for the favorable evaluation of this work’s significance.

Audience: evolutionary biologists, reproductive biologists, *Drosophila* geneticists, and cell biologists (among others) will be interested in this paper.My expertise is in the first three areas just noted.Reviewer #3 (Evidence, reproducibility and clarity (Required)):Summary:The manuscript describes the role of atypical actin related protein Arp53D in *Drosophila* spermatogenesis and reproductive fitness. The gene is highly conserved and expresses at a relatively higher level in the male gonad. Phylogenetic analysis suggested that the gene enveloped relatively late during the specification of Drosophila genus. The authors show that Arp53D knock-out is homozygous viable and mutant males have enhanced fertility. Although the protein localizes at the fusome and investment cone during sperm development, it appeared to be totally redundant for spermatogenesis. Therefore, to understand the reason for its conservation, the authors estimated the maternal and zygotic roles of Arp53D during embryonic development. Arp53D mRNA is enriched in the early stage embryos and the viability of the flies lacking the maternal supply of the protein are susceptible to heat stress at 29C during embryonic development. Thus, the authors conclude that maternal Arp53D is essential to safeguard gastrulation under heat stress, which is the reason for its conservation. The data presentation is top class, experiments are rigorously performed with all appropriate controls, and the reagents are fully characterized. Overall, it is a solid piece of work which raised several intriguing questions.

We thank the reviewer for the positive comments about our work and its impact.

Major Comment:1) The manuscript lacks focus. It appeared to address two totally disconnected question -a) what is the role of Arp53D in sperm development and b) why it is conserved in *Drosophila?* It provides and extensive cell biology characterization of Arp53D localization during spermatid development and a rigorous genetic analysis of the fertility and survival phenotype. However, the causal link between these two data sets are rather tenuous. Hence, the first part of the manuscript is disconnected form the second part. The cause of the enhanced fertility of Arp53D-KO males remained unclear. Similarly, the reason behind reduced survival under heat stress and maternal input of Arp53D has not been investigated.

We appreciate the reviewer’s concern. Although it would appear that this seems like a two-part story, in fact it accurately portrays the logical progression in our functional dissection of Arp53D. For example, its high testis-specific expression logically led us to cytological characterization in the testes, while our unexpected finding that genetic ablation leads to increased male fertility eventually led to our discovery of a maternal-zygotic role for this protein in embryonic viability and development. Furthermore, our cytological characterization in the testes also allowed us to identify the importance of Arp53D’s unusual N-terminal tail in functional diversification.

We also agree with the reviewer’s criticism that the nature of enhanced fertility and reduced survival was not fully elucidated in the previous version. We have now added substantial new experiments for both of these. We show that the increased male fertility of *Arp53D*-KO males may be due to accelerated sperm production (Figure 5C-D), whereas the decreased embryonic survival of Arp53D-KO embryos upon heat stress is correlated with an increase in nuclear abnormalities and a ‘nuclear fallout phenotype,’ in which more damaged nuclei are removed from the *Drosophila* epithelium in *Arp53D*-KO than in WT embryos (Figure 9B-E). The molecular dissection of these traits is of high interest to us, but beyond the scope of the present study.

2) Arp53D protein is absent from the ovary and the mRNA is present in early embryo. Its unclear why the author did not look for the presence of Arp53D mRNA in ovary and the protein in the embryo. The RT-PCR data of the whole fly could be misleading.

We agree with these comments. In our revision, we now include RT-PCR analyses on dissected ovaries to show unambiguous expression of *Arp53D* in ovaries (Supplementary Figure S7B), consistent with published RNA-seq data. Our efforts at characterizing sfGFP-Arp53D in early embryos have been stymied due to high levels of autofluorescence and current lack of information about the developmental stage at which Arp53D protein is expressed.

3) What is the underlying cause of enhanced fertility – increased sperm production of fertilization efficiency? An in depth analysis of the Arp53D-KO phenotype in the testis in terms of the progression of investment cones or the number of intact nuclei bundle at the base of the testis could help to highlight the role of Arp53D in spermatogenesis. Similarly, sperm motility/total sperm estimate in spermatheca may help to suggest whether the mutant sperm has higher fidelity.

We agree and have now performed further cytological analyses of the *Arp53D*-KO testis. We found that *Arp53D* KOs do not appear to have significantly more sperm in the seminal vesicle than WT flies, inferred by measuring the size of the seminal vesicle (Supplementary Figure S5F-G). However, KO testes exhibit more cysts that are undergoing individualization than WT testes, suggesting KO sperm production is accelerated (Figure 5C-D) and likely increases male fertility.

4) The role of maternal and zygotic Arp53D in embryonic development needs to be investigated.

For our revision, we have conducted a cytological analysis of the KO embryos to identify an increased degree of nuclear damage. A full mechanistic understanding of these defects will require new tools and will be a substantial undertaking. We hope the reviewer agrees this is beyond the scope of the present study.

5) Transgenic rescue of the phenotypes described in Figure 5 and 6 is essential. How do the author rule out loss of neighboring gene functions in the KO flies? Arp53D coding region is close to the SOD2 5' and an insertion/deletion of this region may disrupt SOD2 expression.

We agree with this comment and have made substantial efforts to demonstrate transgenic rescue; see response to Reviewer 1 major point 1. We note that *SOD2* expression is not disrupted upon deletion of the *Arp53D* locus (see Supplemental Figure S4D). Moreover, SOD2 is not required for embryonic development but only for adult survival (Mukherjee *et al.*, 2011).

Minor comment:1) The fusome on elongated spermatids morphs into a spectrin cap and promotes membrane extension. Does Arp53D enrichment continue at this stage?

Arp53D enrichment does appear to be stably maintained at spectrin when spermatids are fully elongated.

2) The pictures of intact IC in Arp53D-KO testis would be useful to highlight the role of the protein or the lack of it in sperm-individualization.

We agree. We have imaged actin cones and have found that the KO has significantly more actin cones than WT. We have included this data in Figure 5C-D.

3) The introduction could revised to focus on the competition between male fidelity and embryonic development in the gene evolution.

We appreciate the suggestion. Since we do not have a priori information implicating Arp53D in embryonic development, we expand on this more in the revised Discussion.

4) What is the basis of the term 'rapidly evolving' used in the title. The data suggest that Arp53D is *Drosophila* clade-specific gene.

Based on another reviewer comment, we have revised the title for the paper, which no longer includes ‘rapidly-evolving.’ This term comes from our implementation of the McDonald-Kreitman test, which indicates that Arp53D is evolving under positive selection in *D. melanogaster* (Supplementary Figure S1C).

Reviewer #3 (Significance (Required)):Significance:One of the strongest part of this manuscript is the attempt to correlate the cell biology of actin polymerization with evolution. One of the most exciting proposition highlighted by this manuscript – Apr53D provides an evolutionary fitness by protecting embryonic development under stress.

We thank the reviewer for this positive evaluation, and we agree the strength of the work lies in the evolutionary perspective of Arp53D and how diversification of actin has led to novel roles in development. We hope that our additional experimental analyses have suitably addressed their criticisms.

Existing knowledge:No such literature on Arp53D is found in PubMed.Audience:The manuscript will be appreciated by the evolutionary biologists for its unusual linkage to Actin related protein, the peculiar adverse effect on male fertility, and supportive role in embryonic development.

We thank the reviewer for this positive summary.

Reviewer Competence:This reviewer has first-hand expertise on the cell biology of *Drosophila* spermatogenesis, actin dynamics and embryonic development. The knowledge of the reproductive fitness is secondary, derived from the literature survey.